# *DeTaCH*: DECOUPLING TASKS AND CONTROL VIA A META-GRADIENT HYPERNETWORK

## ABSTRACT

Current language-conditioned robotic policies suffer from a fundamental architectural bottleneck: when language instructions and visual observations are processed through shared representations, networks cannot distinguish between task specification and state perception, leading to policies that exploit spurious visual correlations rather than grounded language semantics. We identify this phenomenon as modality confounding, where gradient interference and entangled representations prevent proper decomposition of task knowledge from perceptual processing. To address this limitation, we propose *DeTaCH*, which reconceptualizes language not as an input to be fused with vision (state), but as a meta-specification that generates parameters of task-specific visuomotor policies. Through a two-stage hypernetwork architecture combining semantic initialization with iterative neural gradient estimation, *DeTaCH* achieves explicit decoupling between language understanding and visual control. Experiments across 90 language-conditioned tasks in LIBERO and 45 tasks in Meta-World demonstrate that *DeTaCH* significantly outperforms state-of-the-art baselines (e.g., Octo), improving absolute success rates from 46.1% to 51.4% (+5.3%) on LIBERO and from 90.6% to 92.2% (+1.6%) on Meta-World. Notably, we observe particularly strong gains on complex, long-horizon tasks where modality confounding is most severe. The generated parameter manifold also exhibits semantic structure, enabling 25% better few-shot adaptation than baselines with only three demonstrations. Our results suggest that explicit architectural separation of heterogeneous modalities may be essential for the generalization of multi-task manipulation policies.

## 1 INTRODUCTION

Language-conditioned robotic manipulation requires policies to simultaneously process linguistic instructions and visual observations, yet current architectures fundamentally fail to maintain the distinction between task specification and state perception. When neural networks process language and vision through shared representations, they face conflicting optimization objectives: learning invariances for language understanding (where "pick the red block" and "grasp the crimson cube" should map similarly) while maintaining sensitivity to visual distinctions (where subtle position differences determine actions). This is also true when learning invariances for visual understanding. This forced coupling through shared parameters creates a phenomenon we term modality confounding, where gradients from visual prediction interfere with those from language understanding. The consequences extend beyond performance degradation: these architectures lose the compositional structure inherent in language and fail to leverage the natural decomposition between task knowledge and perceptual processing.

The inability to properly decouple task specification from state observation creates critical scaling barriers for practical deployment. When current architectures receive modified instructions – such as changing sorting criteria in manufacturing – the entangled representations require restructuring the entire feature space, potentially corrupting previously learned visual skills. This architectural limitation further compounds with task complexity, as interference between modalities grows multiplicatively with more objects, spatial relations, and sequential steps.

The technical challenges underlying modality confounding stem from essential incompatibilities in how language and vision must be processed for effective robot control. Language requires learning

abstract compositional rules – recognizing that "pick" and "grasp" denote similar actions regardless of visual context – while visual processing demands precise spatial reasoning where millimeter differences determine grasp success. When these modalities share parameters, gradient updates that improve visual prediction could often disrupt linguistic structures, creating optimization interference where dense visual feedback may dominate sparse language supervision. Furthermore, language specifies the entire task upfront while visual observations evolve continuously, forcing shared representations to simultaneously encode static task definitions and dynamic state information.

Recent approaches to language-conditioned manipulation have employed various architectural strategies, yet each still faces the issue of modality entanglement, which can manifest as competition between textual and visual modalities (Tang et al., 2025; Liu et al., 2025), neglect of a particular modality such as vision (Mullick et al., 2025), or failure to attend to the correct visual entities (Pani & Yang, 2025; Alonso et al., 2025), all of which often hinder the performance of VLMs or VLAs. Transformer-based methods concatenate language and visual tokens for joint processing, allowing arbitrary cross-modal interactions that blur task specification with state observation. Diffusion-based approaches incorporate language through cross-attention or FiLM conditioning (Perez et al., 2018), creating bidirectional dependencies between modalities. In this paper, we propose *DeTaCH*, which fundamentally reconceptualizes the architecture by treating language not as an input to be fused with vision, but as a meta-specification that generates the parameters of a task-specific visuomotor policy. By explicitly decoupling task specification (processed by a hypernetwork) from state observation (processed by the generated policy), *DeTaCH* effectively prevents modality interference at the architectural level.

Specifically, *DeTaCH* achieves state-of-the-art performance across multiple manipulation benchmarks through a carefully designed two-stage hypernetwork architecture that separates language processing from visuomotor control. Our approach first generates semantically-informed policy parameters through a Weight Initialization Network, then refines them using iterative neural gradient estimation that learns task-specific optimization trajectories in parameter space. This explicit decoupling enables superior generalization – *DeTaCH* maintains robust performance under linguistic paraphrasing and achieves better few-shot adaptation with minimal demonstrations, demonstrating that the generated parameter manifold captures meaningful semantic structure. Empirical validation shows consistent improvements, with the advantage increasing on complex, long-horizon tasks where modality confounding is most severe. While our approach generates task-specific parameters at test time, the resulting target policies are compact and execute efficiently, with the explicit decomposition enabling robust generalization that entangled computation cannot achieve.

*In summary,* we make the following contributions: **1)** We identify and formalize modality confounding as a fundamental architectural limitation where shared representations prevent proper decomposition of task specification and state observation. **2)** We propose *DeTaCH*, a novel architecture that explicitly decouples language and vision through a two-stage hypernetwork that generates task-specific visuomotor policies from language instructions. **3)** We demonstrate that explicit decoupling enables superior generalization, linguistic robustness, and few-shot adaptation through extensive experiments on diverse manipulation benchmarks. **4)** We provide analysis showing that *DeTaCH* learns a semantically-structured parameter manifold where related tasks cluster meaningfully, explaining its superior transfer capabilities.

## 2 RELATED WORK

**End-to-End Visuomotor Policies.** Recent progress in language-conditioned robotics has been dominated by large-scale, end-to-end models (Brohan et al., 2022; Zitkovich et al., 2023; Kim et al., 2024; Black et al., 2024; Liu et al., 2024) that formulate visuomotor control as a sequence modeling problem (Chen et al., 2021; Janner et al., 2021). With the capacity of the Transformer architecture (Vaswani et al., 2017), this paradigm usually tokenizes and concatenates multimodal inputs into unified sequences, and achieves success in vision applications (Dosovitskiy et al., 2020; Oquab et al., 2023; Radford et al., 2021). Influential works including RT-1 (Brohan et al., 2022), Gato (Reed et al., 2022), Octo (Team et al., 2024), and VQ-BeT (Shafiullah et al., 2022) exemplify this approach by processing interleaved sequences of image tokens, language instruction embeddings, and proprioceptive states to predict actions. While these models leverage large-scale datasets to achieve impressive generalization through unified representation learning, their architectural choices inher-

ently entangle task specification (language) with state representation (vision). This coupling creates a fundamental bottleneck that risks modality confounding and imprecise instruction grounding.

**Generative Models for Control.** In parallel, generative models – particularly Denoising Diffusion Probabilistic Models (DDPMs) (Ho et al., 2020; Song et al., 2020) – have emerged as a powerful alternative for learning complex robot behaviors. Diffusion Policy (Chi et al., 2023) pioneered the framing of policy learning as conditional denoising, generating smooth action trajectories from expert demonstrations. These models employ conditioning mechanisms to guide generation based on visual and proprioceptive inputs, often implemented through FiLM layers (Perez et al., 2018) or cross-attention, with architectures having evolved from U-Nets (Ronneberger et al., 2015) to Transformer-based backbones such as DiT (Peebles & Xie, 2023). While these methods excel at generating high-fidelity trajectories, their reliance on conditioning within the generative process still results in task-state entanglement and modality confounding, where low-dimensional language gradients interfere with high-dimensional visual gradients.

**Hypernetworks for Policy Generation.** An alternative paradigm originates from meta-learning (Finn et al., 2017; Snell et al., 2017; Santoro et al., 2016) generates task-specific policy parameters rather than learning a single universal policy. This approach employs hypernetworks (Ha et al., 2016; Beck et al., 2023; Huang et al., 2021), where one network is trained to output the weights of another. This concept has been explored in robotics for multi-task policy generation, with methods such as HyPoGen (Ren et al., 2025) and HyperZero (Rezaei-Shoshtari et al., 2023) demonstrating the ability to generate policies from latent task embeddings. While they can be conceptually used with our approach, their hypernetwork architectures face a critical bottleneck: they typically employ simple MLP-based generators, with limited capability, that struggle to produce full parameterizations of competent, high-performance policies.

## 3 METHOD

**Problem Formulation.** We aim for multi-task robotic manipulation, where an agent must learn to perform diverse tasks specified by natural language instructions. Generally, this can be formalized as a language-conditioned Markov Decision Process (MDP), described by the tuple $\mathcal{M} = (\mathcal{S}, \mathcal{A}, \mathcal{T}, R, \gamma, \mathcal{J})$, where $\mathcal{S} \subseteq \mathbb{R}^{H \times W \times C} \times \mathbb{R}^{d_s}$ represents the state space combining visual and proprioceptive information, $\mathcal{A} \subseteq \mathbb{R}^{d_a}$ is the continuous action space, $\mathcal{T} : \mathcal{S} \times \mathcal{A} \to \Delta(\mathcal{S})$ defines the transition dynamics, $R : \mathcal{S} \times \mathcal{A} \to \mathbb{R}$ is the reward function, $\gamma \in [0, 1)$ is the discount factor, and $\mathcal{J}$ is the space of natural language instructions.

Usually, due to the lack of quality reward functions, we employ behavior cloning to learn a policy $\pi$. Given a dataset $\mathcal{D} = \{(\tau_i, \mathcal{D}_i)\}_{i=1}^N$ containing $N$ distinct tasks, where each task $\tau_i$ is specified by instruction $l_i \in \mathcal{J}$ and has $n_i$ expert demonstrations $\mathcal{D}_i = \{\xi_j^i\}_{j=1}^{n_i}$. Each trajectory $\xi_j^i = \{(o_t^{i,j}, a_t^{i,j})\}_{t=0}^{T_j^i}$ consists of observation-action pairs, where $o_t^{i,j} = (I_t^{i,j}, s_t^{i,j})$ contains an RGB image and proprioceptive state (see Appendix A.2 for detailed specifications). The behavior cloning (BC) objective minimizes:

$$\mathcal{L}_{\text{BC}}(\theta) = \mathbb{E}_{i \sim [N], j \sim [n_i], t \sim [T_j^i]} \left[ \|\pi(o_t^{i,j}, l_i; \theta) - a_t^{i,j}\|_2^2 \right] \tag{1}$$

As shown, current approaches typically learn a monolithic policy $\pi(o, l; \theta) \to a$ that fuses language (task) and visual inputs (state) through shared representations, *therefore,* creating a fundamental tension: the network must simultaneously parse linguistic semantics and interpret visual states within an entangled feature space. Consequently, policies could exploit spurious visual correlations for task completion rather than properly grounding language instructions, as elaborated in the following.

### 3.1 LIMITATIONS OF TASK-STATE ENTANGLEMENT

Although recent Transformer- and diffusion-based architectures have achieved impressive results in language-conditioned robotic manipulation, they share a critical architectural limitation: *Task-State Entanglement* – the practice of processing language and vision through shared, entangled representations. We identify this as a major bottleneck preventing robust language grounding in current approaches.

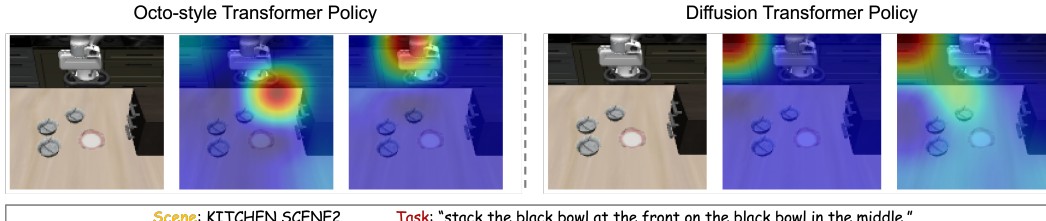

Figure 1: **Evidence of Modality Confounding.** Attention maps from state-of-the-art policies reveal systematic failure to ground language instructions due to task-state entanglement. This reliance on general visual patterns over language semantics evidences modality confounding, where the network cannot effectively disentangle task specification from visual state representation.

Moreover, we propose that the entanglement of task and state could lead to **modality confounding**, a phenomenon where policies may struggle to maintain a clear boundary between task specifications that define the task (e.g., the instruction) from the current state that directly constrains the prediction of action. Consider a policy that must map observations $o$ and language instructions $l$ to actions $a$. In task-state entangled architectures, language and vision are processed through shared representations (see Appendix A.3 for specific architectural examples). For example, transformer-based policies concatenate language tokens with visual patches:

$$\text{Output} = \text{Transformer}(\text{Concat}(T_l, T_v); \theta) \tag{2}$$

where $T_l$ and $T_v$ are language and visual tokens. Similarly, Diffusion Policies condition their action denoising process using cross-attention or FiLM layers:

$$a_t = \text{Denoise}(\epsilon_t | e_l, e_v; \theta) \tag{3}$$

where $e_l$ and $e_v$ are language and visual embeddings, and $\epsilon_t$ is the noisy action. In both cases, a single network is enforced to simultaneously interpret visual states and parse linguistic commands from a mixed-feature space, creating a challenge that may induce the network to leverage visual patterns rather than language specification for task completion.

**Empirical Evidence.** Figure 1 provides examples of the effect of modality confounding in state-of-the-art policies. When given specific language commands (e.g., "stack the black bowl" or "pick up the alphabet soup"), attention visualizations reveal that models consistently focus on the robot's end-effector or background regions rather than the objects specified in the instructions. This systematic misalignment between linguistic instructions and visual attention demonstrates that task-state entangled architectures tend to exploit short-cut visual patterns rather than grounding language semantics to relevant visual entities for accomplishing the manipulation task.

To rigorously diagnose this phenomenon, we employed standard patch-wise attention analysis: visualizing aggregated image-to-language attention weights for Transformers and image-to-action cross-attention for Diffusion models (see Appendix A.10 for rigorous definitions). While recent studies in human-robot interaction suggest that aligning attention with task-relevant objects is critical for task performance (Pani & Yang, 2025), our baselines frequently exhibit a fixation on the end-effector. We attribute this to *shortcut learning* (Xing et al., 2025): entangled architectures exploit spurious correlations (e.g., gripper position) rather than learning grounded semantics. While this heuristic yields competitive performance on simple tasks (e.g., Octo's performance on short-horizon tasks), our experiments show it leads to brittleness in complex, long-horizon scenarios where robust semantic understanding is required.

**Consequences for Policy Generalization.** Modality confounding could severely limit the generalization capabilities of task-state entangled architectures in the following aspects:

- **Brittleness to instruction variations:** Policies could fail when given semantically equivalent but syntactically different instructions, i.e., they may not learn true language-to-action mappings but rather dataset-specific patterns.

- **Poor compositional generalization:** Networks may not be able to recombine learned concepts (e.g., "pick" and "red block") for novel instructions ("pick the red block") since concepts are entangled with specific visual contexts from training.

- **Spurious correlation dependence:** Policies could break when visual contexts change (new backgrounds, lighting, or camera angles), due to their reliance on incidental visual features rather than language understanding.

*Therefore,* we propose to treat language not as an input feature to be fused, but as a *configuration signal* that generates the parameters of a task-specific visuomotor policy. This explicit decoupling of task from state prevents modality confounding at the architectural level and shall lead to more efficient policies.

### 3.2 *DeTaCH*: Explicit Modality Decoupling via Policy Generation

To overcome the fundamental limitations of task–state entanglement, we propose a paradigm shift: instead of learning a monolithic policy that processes fused multi-modal representations, we introduce *DeTaCH*, which treats language as a *structuring signal* that configures the parameters of a task-specific visuomotor policy via a hypernetwork. Architecturally, this hypernetwork employs an iterative refinement design that is reminiscent of inner-loop updates in meta-learning, but this similarity is purely structural: *DeTaCH* is trained end-to-end with standard supervised behavior cloning, without any bi-level optimization or task-specific gradient-based adaptation. As a result, our framework enjoys the representational benefits of meta-inspired refinement while avoiding the training instabilities typically associated with meta-learning methods.

More explicitly, instead of learning $\pi(o, l; \theta) \rightarrow a$ with entangled modalities, *DeTaCH* decomposes the problem into two explicit stages: (1) Language generates task-specific policy parameters: $l \rightarrow \theta_\pi$, and (2) The generated policy performs pure visuomotor control: $\pi(o; \theta_\pi) \rightarrow a$. This architectural separation ensures language and vision never mix in shared representations, effectively preventing modality confounding.

Specifically, *DeTaCH* achieves explicit task-state disentanglement through two distinct components (illustrated in Figure 2):

**1. Hypernetwork** $\mathcal{H}_\phi : \mathbb{R}^{d_l} \rightarrow \Theta$ takes a language embedding and generates policy parameters. Given instruction $l \in \mathcal{J}$, we first encode it using a frozen language encoder $\Phi_L$ to obtain task embedding $e_l \in \mathbb{R}^{d_l}$:

$$e_l = \Phi_L(l) \tag{4}$$

The hypernetwork then maps this embedding to a complete set of policy parameters:

$$\theta_\pi = \mathcal{H}_\phi(e_l) \tag{5}$$

**2. Target Policy** $\pi_{\theta_\pi} : \mathcal{S} \rightarrow \mathcal{A}$ is a task-specific visuomotor controller parameterized by the generated weights $\theta_\pi$. It maps observations directly to actions without any language input:

$$a_t = \pi(o_t; \theta_\pi) \tag{6}$$

This formulation fundamentally decouples the (high-level) knowledge of *how to perform a task* (encoded in the language instruction and embodied in the generated policy parameters $\theta_\pi$) from the (low-level) observation of *what the current state is* (captured by $o_t$). Instead of learning a general mapping $(l, o_t) \rightarrow a_t$, our method learns a compositional process: $l \rightarrow \theta_\pi$ followed by $(o_t, \theta_\pi) \rightarrow a_t$. By generating a specialized policy for each task, the hypernetwork is compelled to grasp the underlying structure of the task manifold, promoting generalization and data efficiency.

**Hypernetwork Architecture – The Key to Effective Decoupling.** While the above decomposition conceptually addresses modality confounding, its practical success critically depends on the hypernetwork's ability to generate high-quality, task-specific parameters. Directly generating a vast number of parameters from a language embedding is an ill-posed regression problem that would fail without careful architectural design. Our key technical contribution is a sophisticated two-stage hypernetwork that makes this challenging parameter generation tractable and effective.

More explicitly, the naive approach of directly mapping $e_l \rightarrow \theta_\pi$ faces two fundamental challenges: (1) the vast dimensionality of modern policy networks makes direct regression intractable, and (2) the highly non-linear relationship between language semantics and optimal policy parameters requires sophisticated intermediate computations. To address these challenges, *DeTaCH* employs a

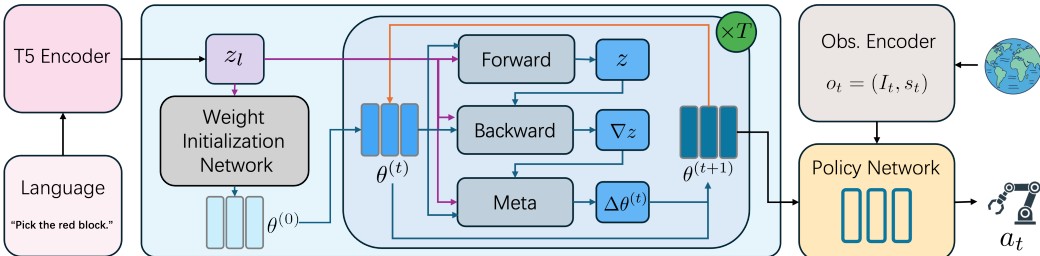

Figure 2: **The *DeTaCH* architecture.** Language instruction $l$ is encoded into embedding $e_l$ and processed by the hypernetwork through two stages: (1) Weight Initialization Network generates initial parameters $\theta_\pi^0$, and (2) Iterative Refinement module updates parameters over $T$ steps to produce final $\theta_\pi$. The generated parameters configure the target policy $\pi_{\theta_\pi}$, a task-specific visuomotor controller. Note that state observations $o_t$ only enter at the target policy stage, maintaining complete separation from language processing. This explicit decoupling prevents modality entanglement by treating language as a meta-specification for policy generation rather than a common input to be fused with observed states.

carefully designed coarse-to-fine generation strategy that transforms the problem from direct regression to iterative refinement:

**Stage 1: Weight Initialization Network (WIN).** The WIN, parameterized by $\phi_1$, generates an initial set of policy parameters from the language embedding:

$$\theta_\pi^0 = \text{WIN}_{\phi_1}(e_l) \tag{7}$$

This provides a semantically-informed initialization that positions the parameters in a promising region of the vast parameter space. However, this coarse initialization alone is insufficient for high-performance policies.

**Stage 2: Iterative Parameter Refinement.** The critical innovation is our iterative refinement module, which transforms the coarse initialization into task-optimized parameters. Rather than attempting to predict perfect parameters in one shot, this module learns to simulate an optimization process tailored to each task. The refinement module, parameterized by $\phi_2$, performs $T$ update steps:

$$\theta_\pi^{t+1} = \theta_\pi^t + \Delta\theta^t, \quad \text{where} \quad \Delta\theta^t = f_{\phi_2}(\theta_\pi^t, e_l) \tag{8}$$

This module learns task-specific update rules that progressively optimize the parameters. Crucially, it does not perform standard gradient descent but instead learns a neural optimization trajectory through parameter space (see Appendix A.4 for more architectural design details). The refinement process allows the hypernetwork to make fine-grained adjustments that would be impossible to achieve through direct regression, enabling the generation of high-quality, task-specialized policies.

The entire framework is trained end-to-end by optimizing $\phi = \{\phi_1, \phi_2\}$ using the behavior cloning loss from Equation 1, where the target policy with generated parameters $\theta_\pi = \theta_\pi^T$ is applied to mimic the demonstration data.

### 3.3 FEW-SHOT ADAPTATION AS AN EMERGENT META-LEARNING CAPABILITY

A key advantage of *DeTaCH*'s explicit modality decoupling is its natural ability to adapt to novel tasks from minimal demonstrations. By separating task-agnostic meta-knowledge (encoded in the hypernetwork $\mathcal{H}_\phi$) from task-specific parameters (the generated policy $\theta_\pi$), our architecture exhibits emergent meta-learning capabilities without requiring specialized meta-training procedures.

**Meta-Learning Perspective.** Specifically, the hypernetwork $\mathcal{H}_\phi$ functions as a meta-learner that captures the underlying structure of the task manifold rather than memorizing individual task solutions. During multi-task training, $\mathcal{H}_\phi$ learns to encode abstract compositional principles – such as the kinematic invariances between semantically related actions – into its parameterization $\phi$. This learned mapping from semantic space to parameter space represents a form of meta-knowledge: an understanding of how language concepts translate to visuomotor control structures. In contrast, monolithic policies with task-state entanglement conflate multiple levels of abstraction within a single entangled parameter space, precluding efficient task-specific adaptation (see Appendix A.5 for detailed comparisons with parameter-efficient fine-tuning methods).

**Efficient Test-Time Adaptation.** When presented with a novel task $\tau_{\text{new}}$ with instruction $l_{\text{new}}$ and a limited support set $\mathcal{D}_{\text{new}} = \{\xi_j\}_{j=1}^K$, *DeTaCH* performs rapid adaptation through a two-stage process:

**1. Task-Informed Initialization:** The frozen hypernetwork generates initial parameters conditioned on the new task's semantic representation:

$$\theta_\pi^{\text{init}} = \mathcal{H}_\phi(\Phi_L(l_{\text{new}})) \tag{9}$$

This initialization exploits the learned task manifold structure to position the parameters in a semantically-appropriate region of the parameter space, providing significantly better conditioning than random initialization.

**2. Targeted Parameter Optimization:** Only the generated target policy parameters undergo gradient-based optimization on the demonstration data:

$$\theta_\pi^* = \arg\min_{\theta_\pi} \sum_{\xi_j \in \mathcal{D}_{\text{new}}} \sum_{(o_t, a_t) \sim \xi_j} \|\pi(o_t; \theta_\pi) - a_t\|_2^2, \quad \text{initialized at } \theta_\pi^{\text{init}} \tag{10}$$

This adaptation mechanism achieves superior sample efficiency compared to fine-tuning entangled architectures. The hypernetwork-provided initialization constrains the optimization to a well-conditioned subspace of the parameter manifold, transforming adaptation into a local refinement problem rather than a global search. The frozen hypernetwork preserves its meta-knowledge throughout adaptation, preventing catastrophic forgetting of the learned task structure (implementation details in Appendix A.5).

## 4 EXPERIMENT

We conduct experiments to validate that explicit modality decoupling through *DeTaCH* improves language-conditioned robotic manipulation. Our evaluation addresses three key research questions:

1. **Multi-Task Performance (RQ1):** Does *DeTaCH*'s explicit decoupling architecture outperform state-of-the-art task-state entangled methods on diverse manipulation tasks?
2. **Few-Shot Adaptation (RQ2):** Can the structured parameter manifold learned by *DeTaCH* enable more efficient adaptation to novel tasks compared to entangled representations?
3. **Architectural Analysis (RQ3):** How does our proposed architecture improve robustness to language variations and learn a semantically structured parameter space?

**Experimental Overview.** We evaluate on two benchmarks: LIBERO-90 (90 language-conditioned tasks) and Meta-World ML45 (45 goal-conditioned tasks). Section 4.1 details our experimental setup. Sections 4.2 and 4.3 present multi-task and few-shot results respectively. Section 4.4 provides ablations and qualitative analysis. Appendix A.9.5 shows the result on real robots.

### 4.1 EXPERIMENTAL SETUP

**Benchmarks.** We evaluate our method on two challenging multi-task manipulation benchmarks: LIBERO-90 (Liu et al., 2023) and Meta-World (Yu et al., 2020). For LIBERO, we use 80 of its 90 language-conditioned tasks for training, with the remaining 10 held out for few-shot adaptation. For Meta-World, we adopt the ML45 protocol, using all 45 goal-conditioned tasks for training and holding out 5 for evaluation. We use the 50 human demonstrations provided for each LIBERO task and collect 100 expert demonstrations for each Meta-World task. Further details on the specific task splits, categories, and benchmark characteristics are deferred to Appendix A.6.1.

**Baselines** We compare our method against six state-of-the-art baselines, which are divided into two categories. The first category, **Monolithic Methods**, includes Transformer-based approaches such as Octo (Team et al., 2024) and VQ-BeT (Shafiullah et al., 2022), alongside diffusion-based models like Diffusion Policy (Chi et al., 2023) and DiT (Chi et al., 2023). The second category, **Hypernetwork-based Methods**, consists of HyPoGen (Ren et al., 2025) and HyperZero (Rezaei-Shoshtari et al., 2023).

We do not include Pi0 (Black et al., 2024) or OpenVLA (Kim et al., 2024) families in our comparison. This decision is based on our desire to ensure an apples-to-apples comparison, as these

Table 1: Multi-task performance on LIBERO-90. Success rates (%) averaged over 50 episodes per task with novel initial configurations. *DeTaCH* shows increasing advantages on complex tasks where modality confounding is most problematic. Best in **bold**, second-best underlined.

| Task Category | Octo | VQ-BeT | Diff. | DiT | H-Zero | H-Gen | *DeTaCH* |
|---|---|---|---|---|---|---|---|
| LIBERO-easy | 65.0 | **72.0** | 37.2 | 20.3 | 15.8 | 32.0 | 63.7 |
| LIBERO-medium | 51.2 | 35.7 | 32.9 | 12.6 | 28.6 | 36.5 | **58.6** |
| LIBERO-long | 38.3 | 19.2 | 22.7 | 9.7 | 20.4 | 31.5 | **43.2** |
| Overall | 46.1 | 31.1 | 28.2 | 11.9 | 23.3 | 33.6 | **51.4** |

methods are significantly larger in parameter size (approximately 100x larger) and are pre-trained on very large, diverse datasets. These factors introduce an additional axis of variation, making it difficult to isolate the architectural differences that are central to our study. To maintain fairness, we focus on models that are trained from scratch using identical visual (ResNet-18) and language (T5-small) encoders. Detailed descriptions of each baseline's architecture and our implementation specifics are deferred to Appendix A.7.2.

**Training Protocol.** Across all experiments, we follow the behavior cloning paradigm. The total training dataset consists of $50 \times 80 = 4000$ demonstrations for LIBERO and $100 \times 45 = 4500$ demonstrations for Meta-World. Each timestep provides a front-view RGB image (resized to $128 \times 128$), the robot's proprioceptive state, and the corresponding ground-truth action. All models are trained end-to-end for 200 epochs using the AdamW optimizer with a batch size of 256. We employ a cosine learning rate schedule with an initial learning rate of $1 \times 10^{-4}$. Detailed implementation specifics are available in the Appendix A.9.1. To isolate the impact of model architecture from other engineering factors, we standardize the training protocol across all methods by scaling models to approximately 25M parameters and excluding engineering enhancements such as image augmentation and action smoothing.

However, to provide a more comprehensive evaluation, we also incorporated these engineering enhancements into both DeTaCH and the baselines, specifically replicating the experimental settings of Tables 1, 3, and 4 with these "enhanced versions". We observed that while these techniques yield significant performance gains across the board, DeTaCH consistently outperforms the baselines in both the standardized "clean" setting and the "enhanced" setting, particularly reaching as high as 92.8% on the LIBERO-90 benchmark. This bi-directional verification demonstrates the robustness and superiority of our architecture independent of auxiliary engineering tricks. Please refer to Appendix A.9.4 for exact numerical results.

**Evaluation Metrics.** We report task success rates averaged over multiple trials under two distinct evaluation settings. For the multi-task benchmark, we assess intra-task robustness over 50 episodes per task, each with a novel initial configuration. For the few-shot setting, we evaluate adaptation performance over 25 episodes after the model has been fine-tuned on K=3 demonstrations.

### 4.2 MULTI-TASK PERFORMANCE

We evaluate whether *DeTaCH*'s explicit modality decoupling improves generalization within training tasks. Models are tested on the same tasks they were trained on, but with novel initial configurations (unseen object positions, robot poses) to assess robustness.

**Results on LIBERO-90.** Table 1 reveals a critical pattern: while Octo and VQ-BeT compete on simple tasks, *DeTaCH* dominates on complex, long-horizon tasks. For example, the performance gap increases from -8.3% on short/easy tasks to +24.0% on long/complex tasks (vs. VQ-BeT) in absolute points, supporting our hypothesis that task-state entangled architectures suffer from modality confounding in challenging scenarios. *DeTaCH*'s 51.4% overall success rate represents a **5.3% absolute improvement** over the best baseline (Octo).

**Results on Meta-World.** Meta-World results (Table 2) confirm *DeTaCH*'s advantages generalize across benchmarks. While simple `open&close` tasks saturate near 100% for most methods, *DeTaCH* shows clear improvements on tasks requiring precise object manipulation (`pick&place`: +5.0% over Octo) and complex action sequences (`others`: +2.9%). The consistent performance across categories demonstrates that explicit modality decoupling benefits diverse manipulation primitives, not just specific tasks.

Table 2: Multi-task performance on Meta-World ML45 grouped by manipulation primitives. *DeTaCH* maintains consistent advantages across diverse action types, achieving 92.2% overall success.

| Task Category | Octo | VQ-BeT | Diff. | DiT | H-Zero | H-Gen | *DeTaCH* |
|---|---|---|---|---|---|---|---|
| open&close | 99.3 | **100.0** | 56.8 | 52.7 | 99.3 | 91.5 | 99.5 |
| pick&place | 83.0 | 69.0 | 19.5 | 19.0 | 81.5 | 66.5 | **88.0** |
| press&pull | 93.4 | 88.1 | 33.9 | 38.5 | 38.4 | 72.5 | **93.4** |
| others | 85.8 | 77.6 | 54.6 | 48.1 | 48.2 | 68.5 | **88.7** |
| Overall | 90.6 | 84.6 | 44.5 | 42.9 | 56.8 | 73.8 | **92.2** |

Table 3: Few-shot adaptation on held-out tasks. Success rates (%) after 1000 gradient steps with K=3 demonstrations. *DeTaCH*'s structured initialization enables superior adaptation efficiency.

| Dataset | Octo | VQ-BeT | Diff. | DiT | H-Zero | H-Gen | *DeTaCH* | Δ |
|---|---|---|---|---|---|---|---|---|
| LIBERO-90 | 12.0 | 6.8 | 0.4 | 3.2 | 14.4 | 14.0 | **18.0** | +3.6 |
| Meta-World | 27.2 | 32.0 | 11.2 | 12.8 | 26.4 | 26.4 | **34.4** | +2.4 |
| Average | 19.6 | 19.4 | 5.8 | 8.0 | 20.4 | 20.2 | **26.2** | +5.8 |

**Key Finding:** *DeTaCH*'s advantage correlates with task complexity – minimal on simple tasks but substantial on complex ones, validating that modality confounding becomes increasingly problematic as task horizon and complexity grow.

### 4.3 FEW-SHOT ADAPTATION TO UNSEEN TASKS

We evaluate adaptation efficiency on held-out tasks under minimal demonstrations (K=3). This tests whether *DeTaCH*'s structured parameter manifold enables better transfer compared to entangled representations or architectures.

**Adaptation Protocol** Our adaptation protocol is designed for a fair comparison across architectures. Hypernetwork methods first generate a task-specific policy, after which only the generated parameters are fine-tuned. In contrast, task-state entangled baselines are adapted using LoRA (Hu et al., 2022) adapters, with a parameter count (0.57M) matched to that of the generated policies. To prevent knowledge leakage, each adaptation begins from scratch: policies are regenerated for our method, while baselines are reverted to their pre-trained weights with a fresh LoRA adapter. The adaptation process for all methods consists of 1000 gradient steps on 3 demos, and post-adaptation performance is averaged over 25 episodes per task. More details can be found in Appendix A.5.

**Adaptation Results** Table 3 demonstrates *DeTaCH*'s superior adaptation capability. On LIBERO's held-out tasks, *DeTaCH* achieves 18.0% success – a 25% relative improvement over the best baseline (HyperZero, 14.4%). The advantage extends to Meta-World (+2.4% absolute), indicating robust transfer across task distributions. Notably, diffusion-based methods (Diffusion Policy, DiT) catastrophically fail at few-shot adaptation despite reasonable multi-task performance, suggesting their entangled representations may prevent efficient fine-tuning. In contrast, *DeTaCH*'s explicit decoupling provides a well-conditioned optimization landscape: the hypernetwork-generated initialization $\theta_\pi^{\text{init}} = \mathcal{H}_\phi(\Phi_L(l_{\text{new}}))$ positions parameters near task-appropriate optima, requiring only local refinement rather than global search.

### 4.4 ANALYSIS AND ABLATIONS

**Robustness to Language Variations.** We test whether explicit decoupling improves robustness to linguistic variations by training on 50 paraphrased instructions per task (e.g., "pick up the red block" → "grab the crimson cube") and evaluating on 10 additional unseen descriptions. As shown in Table 4, *DeTaCH* achieves the highest performance under linguistic variations. On LIBERO, *DeTaCH* attains 39.8% success rate on paraphrased instructions compared to Octo's 37.2%, and ours has 91.8% success rate compared to Octo's 90.3% on Meta-World. This maintained performance advantage under linguistic variations suggests that task-state decoupled architectures can better capture the underlying semantic structure of instructions rather than overfitting to specific phrasings.

Table 4: Performance with augmented language instructions. *DeTaCH* maintains higher success rates under linguistic variations.

| Dataset | Octo | VQ-BeT | Diff. | DiT | H-Zero | H-Gen | *DeTaCH* |
|---|---|---|---|---|---|---|---|
| LIBERO | 37.2 | 16.8 | 20.9 | 12.5 | 32.9 | 36.1 | **39.8** |
| Meta-World | 90.3 | 85.9 | 49.5 | 23.3 | 79.2 | 83.1 | **91.8** |

**Parameter Space Visualization.** Figure 3 visualizes the parameter manifold learned by *DeTaCH*, showing that the hypernetwork organizes the policy space based on functional and semantic similarity, not just task IDs, via t-SNE (Maaten & Hinton, 2008). Analysis of the clusters reveals three key properties: (1) Scene Invariance: Tasks 1 and 2 (Turn on the stove) cluster together despite being in different scenes, indicating that the hypernetwork focuses on core task semantics while ignoring visual distractors. (2) Functional Grouping: A cohesive "drawer manipulation" region (Tasks 6–11) groups tasks by target object (drawers), maintaining

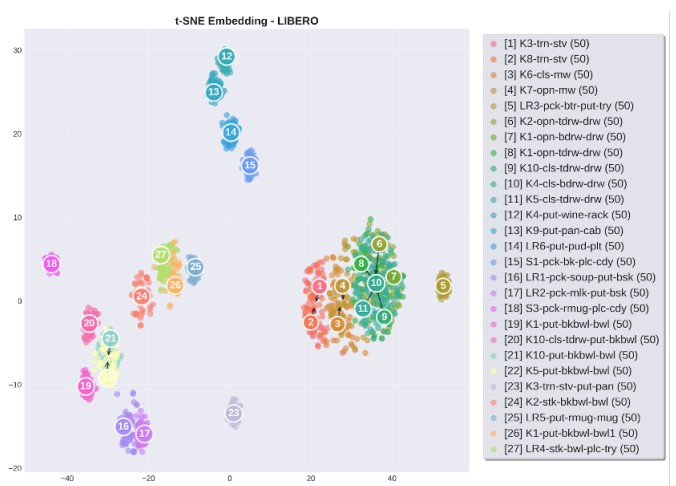

Figure 3: t-SNE visualization of generated parameters. Semantically similar tasks cluster, indicating a structured manifold.

local structure for specific actions. (3) Semantic Continuity: Inverse operations like Open/Close microwave (Tasks 3 and 4) are positioned near each other, suggesting the model captures state-change relationships. This structured organization shows that *DeTaCH* learns a semantic topology, with linguistic concepts mapping to interpolatable regions in the parameter space, supporting superior few-shot adaptation. For detailed task mappings, refer to Appendix A.9.7.

**Summary.** Our ablations confirm that (1) the proposed coarse-to-fine policy generation architecture is essential, (2) explicit task-state decoupling improves linguistic robustness, and (3) the learned parameter space captures semantic task relationships, validating our architectural choices.

## 5 CONCLUSION

We demonstrate that modality confounding fundamentally limits current language-conditioned manipulation approaches. *DeTaCH* addresses this through explicit architectural decoupling, where language generates task-specific policies rather than being fused with visual observations. Our method achieves state-of-the-art performance on LIBERO-90 (51.4%) and Meta-World (92.2%), with performance gaps widening on complex, long-horizon tasks. The emergent semantic structure in the learned parameter manifold enables superior few-shot adaptation, suggesting that proper architectural inductive biases could potentially unlock compositional generalization. Future work could explore whether similar decoupling principles benefit other multi-modal robotic learning paradigms.

**Limitations.** Despite its advantages in inference sample efficiency, DeTaCH has limitations in computational cost and scalability. The iterative refinement module requires storing intermediate computation graphs for gradient estimation, leading to higher training memory overhead compared to standard behavior cloning. Additionally, dynamic parameter generation lacks the low-level hardware optimization found in static architectures like FlashAttention, resulting in lower wall-clock training efficiency. Finally, our implementation is limited to generating weights for MLP policies, and scaling this hypernetwork paradigm to complex topologies, such as deep Vision Transformers, remains a challenge due to the exponential growth of the parameter space.

## 6 ETHICS STATEMENT

This work strictly adheres to the ICLR Code of Ethics. Our research focuses on improving language-conditioned robotic manipulation through explicit modality decoupling and does not involve human subjects, animal experiments, or raise concerns related to privacy, security, or potential harmful deployment. All experiments were conducted in simulation environments using publicly available benchmarks (LIBERO-90 and Meta-World), with demonstrations collected through established protocols. The *DeTaCH* framework is designed for research purposes in controlled robotic manipulation tasks and does not present risks of misuse or harmful applications. We have carefully considered the broader impacts of our work and believe it contributes positively to the advancement of interpretable and robust robotic systems without introducing ethical concerns. The improved sample efficiency and generalization capabilities demonstrated by our method could benefit applications in assistive robotics and industrial automation when properly validated in real-world settings. All authors have thoroughly reviewed and acknowledge compliance with the ICLR Code of Ethics.

## 7 REPRODUCIBILITY STATEMENT

We have taken extensive measures to ensure the reproducibility of our work on the *DeTaCH* framework. The complete architectural details of our two-stage hypernetwork, including the Weight Initialization Network and Iterative Parameter Refinement module with neural gradient estimation, are described in Section 3.2 and Appendix A.4. All experimental configurations, including the task split for LIBERO-90 and Meta-World, batch size, learning rate, and training epochs are detailed in Section 4.1 and Appendix A.6. The behavior cloning loss formulation (Equation 1), hypernetwork parameter generation (Equation 7 and Equation 8), and few-shot adaptation procedure (Algorithm 2, Equation 9, and Equation 10) are precisely specified. Implementation details include the language encoder, visual encoder, and specific hyperparameters for all baselines in Appendix A.7.2. Our ablation studies examining the contributions of each component are documented in Table 6. Code implementation and trained models will be made available upon acceptance to facilitate reproduction of all reported results.

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

# A APPENDIX

## A.1 THE USE OF LARGE LANGUAGE MODELS

Large language models were employed in this work solely as auxiliary tools to aid in the writing and refinement process. Specifically, LLMs assisted with improving clarity of expression, enhancing grammatical correctness, and ensuring consistency in academic writing style across different sections of the paper. All substantive contributions, including the research idea, experimental design, implementation, analysis of results, and core scientific arguments, were conceived and executed entirely by the authors. The LLMs played no role in the formulation of research questions, methodology development, or the generation of any technical content or experimental results. The authors take full responsibility for all content in this paper, including the accuracy of all claims, experimental results, and citations.

## A.2 PROBLEM FORMULATION DETAILS

### A.2.1 STATE AND ACTION SPACES

The state space $\mathcal{S} \subseteq \mathbb{R}^{H \times W \times C} \times \mathbb{R}^{d_s}$ consists of:

- RGB images $I_t \in \mathbb{R}^{H \times W \times C}$ with resolution $H \times W$ and $C = 3$ color channels
- Proprioceptive state $s_t \in \mathbb{R}^{d_s}$ including end-effector pose and gripper state.

The continuous action space $\mathcal{A} \subseteq \mathbb{R}^{d_a}$ represents target joint velocities or end-effector displacements, depending on the control mode. The transition dynamics $\mathcal{T} : \mathcal{S} \times \mathcal{A} \to \Delta(\mathcal{S})$, where $\Delta(\mathcal{S})$ denotes the probability simplex over $\mathcal{S}$, captures the stochastic nature of real-world manipulation.

### A.2.2 DATASET STRUCTURE

Each demonstration trajectory $\xi_j^i$ has horizon $T_j^i$ and consists of:

- Observations $o_t^{i,j} = (I_t^{i,j}, s_t^{i,j})$ combining visual and proprioceptive information
- Expert actions $a_t^{i,j} \in \mathcal{A}$ recorded from human demonstrations or privileged controllers
- Language instruction $l_i$ describing the task goal in natural language

## A.3 EXAMPLES OF TASK-STATE ENTANGLED ARCHITECTURES

Modern language-conditioned robotic policies employ various forms of task-state entanglement:

**Transformer-based Fusion.** Policies like Octo (Team et al., 2024) tokenize both language and visual inputs, concatenating them into a single sequence for processing. The self-attention mechanism allows arbitrary interactions between language and visual tokens, creating entangled representations throughout the network depth.

**Diffusion-based Fusion.** Diffusion Policies (Chi et al., 2023) incorporate language through:

- **FiLM conditioning:** Language embeddings modulate visual features through affine transformations
- **Cross-attention:** Action denoising queries attend to both language and visual keys
- **Concatenation:** Language and visual embeddings are concatenated before denoising

In all cases, the action prediction emerges from representations where language and vision have been deeply intertwined through non-linear transformations, making it impossible to isolate their individual contributions to the final output.

### A.4 HYPERNETWORK IMPLEMENTATION DETAILS

#### A.4.1 LANGUAGE ENCODER

We employ T5-small (Raffel et al., 2020) as our frozen language encoder $\Phi_L$, which maps natural language instructions to 512-dimensional embeddings. The encoder is pre-trained and remains frozen during training to preserve its linguistic knowledge.

#### A.4.2 NEURAL GRADIENT ESTIMATION AND UPDATE MODULE

The iterative refinement module (Stage 2) employs a sophisticated neural gradient estimation mechanism inspired by meta-learning. At each refinement step $t \in \{0, \ldots, T-1\}$, the module simulates an optimization process through three sub-networks:

**Forward Pass Simulation:** The module first simulates a forward pass through the target network to estimate task-specific activation patterns:

$$(z_0, \ldots, z_n) = F_{\text{Forward}}(\theta_\pi^t, e_l; \phi_{2,f}) \tag{11}$$

where $z_i$ represents estimated activations at layer $i$ and $\phi_{2,f}$ are the forward simulator parameters.

**Backward Pass Simulation:** Next, it simulates a backward pass to compute pseudo-gradients:

$$\left( \frac{\partial L}{\partial z_n}, \ldots, \frac{\partial L}{\partial z_1} \right) = F_{\text{Backward}}(\theta_\pi^t, e_l, z_0, \ldots, z_n; \phi_{2,b}) \tag{12}$$

These pseudo-gradients are not true gradients of an explicit loss function but learned signals that guide parameter updates toward task-specific optima.

**Meta-Network Update:** Finally, a meta-network computes the parameter update:

$$\Delta\theta^t = F_{\text{Meta}} \left( \theta_\pi^t, e_l, \frac{\partial L}{\partial z_n}, \ldots, \frac{\partial L}{\partial z_1}; \phi_{2,m} \right) \tag{13}$$

The complete refinement module parameters are $\phi_2 = \{\phi_{2,f}, \phi_{2,b}, \phi_{2,m}\}$. This design allows the hypernetwork to learn task-specific optimization trajectories in parameter space, effectively performing learned optimization tailored to each language instruction.

#### A.4.3 ARCHITECTURAL DESIGN CHOICES

The WIN can be implemented as either a multi-layer perceptron (MLP) or a Transformer, depending on the complexity of the target network. For our experiments, we use a 4-layer transformer with cross attention blocks as WIN. The refinement module typically performs $T = 3$ update steps, balancing computational efficiency with parameter quality.

#### A.4.4 PSEUDOCODE FOR A SINGLE STEP UPDATE

The update process follows a sequence of steps involving tokenization, forward pass computation, backward pass, gradient calculation, and parameter updates. We show the pseudocode for a single step update in Algorithm 1. We use $\tau_i, \omega_i$ to represent the activation and parameter $z_i, \theta_i$ in the latent space.

### A.5 FEW-SHOT ADAPTATION DETAILS

#### A.5.1 COMPARISON WITH PARAMETER-EFFICIENT FINE-TUNING

Traditional parameter-efficient fine-tuning methods such as Low-Rank Adaptation (LoRA) (Hu et al., 2022) introduce trainable low-rank matrices to adapt pre-trained models. When applied to policies with task-state entanglement, these methods face fundamental limitations:

- **Entangled Update Space:** Low-rank updates must navigate through pre-existing entangled representations where language and visual features are already mixed through multiple layers of non-linear transformations.

---

**Algorithm 1** Single Step Update for DeTaCH

---

1: **Input:** Parameters $\theta^{(t-1)}$, task embedding $e_l$
2: **Output:** Updated parameters $\Delta\theta^{(t-1)}$
3: **Tokenize parameters for each layer $i$'s parameter $\theta_i$:**
4:     $\omega_i \leftarrow \text{Tokenize}(\theta_i)$
5: **Compute initial activation $\tau_0$:**
6:     $\tau_0 \leftarrow \text{CrossAttn}(e_l, \omega_i, \text{learnable param})$
7: **Forward Pass Computation Using Cross-Attention:**
8:     $\tau_i \leftarrow \text{CrossAttn}(\omega_i, \tau_{i-1}, \tau_{i-1})$
9: **Backward Pass and Jacobian Estimation:**
10:     $\frac{\partial z_i}{\partial z_{i-1}} \leftarrow \text{CrossAttn}(\omega_i, \tau_{i-1}, \tau_{i-1})$
11:     $\frac{\partial z_i}{\partial \theta_i} \leftarrow \text{CrossAttn}(\tau_{i-1}, \omega_i, \omega_i)$
12: **Gradient Computation:**
13:     $\frac{\partial L}{\partial z_{i-1}} \leftarrow \text{CrossAttn}\left(\frac{\partial L}{\partial z_i}, \frac{\partial z_i}{\partial z_{i-1}}, \frac{\partial z_i}{\partial z_{i-1}}\right)$
14:     $\nabla\omega_i \leftarrow \text{CrossAttn}\left(\frac{\partial L}{\partial z_i}, \frac{\partial z_i}{\partial \theta_i}, \frac{\partial z_i}{\partial \theta_i}\right)$
15: **Decode into $\theta$ space**
16:     $\Delta\theta_i \leftarrow \text{MLP}(\nabla\omega_i)$ and proper reshape

---

- **Interference Effects:** Modifications intended to accommodate new language instructions can inadvertently perturb established visuomotor control pathways, leading to negative transfer.
- **Suboptimal Initialization:** The base model's parameters are optimized for the training task distribution, providing no task-specific initialization for novel instructions.

In contrast, *DeTaCH* operates in a fundamentally different optimization regime:

- **Decoupled Parameter Space:** Updates occur exclusively in the target policy parameters $\theta_\pi$, which represent pure visuomotor mappings without language entanglement.
- **Preserved Meta-Knowledge:** The hypernetwork $\mathcal{H}_\phi$ remains frozen, maintaining its learned understanding of the task manifold structure.
- **Task-Conditioned Initialization:** The hypernetwork provides semantically-informed parameter initialization based on the language instruction, positioning the optimization in a favorable region of parameter space.

### A.5.2 ADAPTATION ALGORITHM IMPLEMENTATION

For few-shot adaptation with $K$ demonstrations, we employ the following procedure:

---

**Algorithm 2** Few-Shot Adaptation for *DeTaCH*

---

**Require:** Novel task instruction $l_{\text{new}}$, demonstrations $\mathcal{D}_{\text{new}} = \{\xi_j\}_{j=1}^K$
**Ensure:** Adapted policy parameters $\theta_\pi^*$
1: Generate initial parameters: $\theta_\pi^{(0)} \leftarrow \mathcal{H}_\phi(\Phi_L(l_{\text{new}}))$
2: Initialize optimizer (e.g., Adam or SGD)
3: **for** epoch $= 1$ to $N_{\text{epochs}}$ **do**
4:     **for** each batch $B \subset \mathcal{D}_{\text{new}}$ **do**
5:         Compute loss: $\mathcal{L} = \sum_{(o_t, a_t) \in B} \|\pi(o_t; \theta_\pi) - a_t\|_2^2$
6:         Update parameters: $\theta_\pi \leftarrow \text{Optimizer}(\nabla_{\theta_\pi}\mathcal{L})$
7:     **end for**
8:     Perform early stopping based on validation loss (if available)
9: **end for**
10: **return** $\theta_\pi^*$

---

The key insight is that the hypernetwork-generated initialization $\theta_\pi^{(0)}$ provides a strong prior for the target task, enabling rapid convergence with minimal data. The hypernetwork parameters $\phi$ remain

frozen throughout this process, preserving the learned meta-knowledge. This procedure typically converges within 50-100 gradient steps for $K \in [1, 5]$ demonstrations.

## A.6 EXPERIMENTAL DETAILS

### A.6.1 LIBERO TASK SPLITS

The 80 training tasks and 10 held-out tasks in LIBERO-90 are divided as follows:

**Training Tasks (80):**

- **Easy (8 tasks):** Single-object manipulation (e.g., "put the bowl in the cabinet")
- **Medium (32 tasks):** Multi-object coordination (e.g., "stack the red block on the blue block")
- **Long (40 tasks):** Sequential multi-step (e.g., "pick all fruits and place them in the basket")

The exact split of tasks is listed in Table 19.

**Held-out Tasks (10):**

- **Spatial (5 tasks):** Familiar goals in unseen scene configurations
- **Goal (5 tasks):** Novel goals with familiar objects and scenes

The exact split of tasks is listed in Table 20.

## A.7 META-WORLD TASK SPLITS

Similarly, we split Meta-World ML45 into 4 categories:

**Training Tasks (45):**

- **open&close (8 tasks)**: Open or close an object (e.g. "window open")
- **pick&place (4 tasks)**: Pick the object up or place it elsewhere (e.g. "pick out of hole")
- **press&pull (16 tasks)**: Press the button or lever (e.g. "button press topdown")
- **others (17 tasks)**: other tasks not in previous categories (e.g. "soccer")

**Held-out Tasks (5):**

These tasks are identical to the Meta-world's standard ML45 evaluation set.

The exact split of tasks is listed in the Table 21.

### A.7.1 SAMPLING STRATEGY

For all the experiments, we first construct a global list of all transitions across the dataset, where each entry is a tuple (task, episode, timestep). At training time, each batch is formed by uniformly sampling batch size transitions from this list.

For language augmentation experiments, we use the same transition-level sampling strategy. In addition, for each sampled transition we randomly draw a paraphrased instruction from that task's pool of 50 paraphrases, so the model sees diverse linguistic realizations of the same underlying task during training.

### A.7.2 BASELINE IMPLEMENTATION DETAILS

All baselines are implemented in PyTorch with the following specifications:

**Shared Components:**

- Visual encoder: ResNet-18 pretrained on ImageNet, fine-tuned during training
- Language encoder: T5-small encoder (Raffel et al., 2020), frozen
- Action space: 6-DoF end-effector delta and 1-dim state of gripper for LIBERO, 3-DOF end-effector delta and 1-dim gripper for Meta-World

Table 5: Computation cost breakdown for *DeTaCH* and baselines.

| Component / Model | Time (ms) | FPS | Note |
|---|---|---|---|
| *DeTaCH* Single Update Step | 11.95 | $\sim 84$ | Cost of one iterative update |
| *DeTaCH* Weight Gen. | 39.65 | $\sim 25$ | Full hypernet run (once per task, 3 updates) |
| *DeTaCH* Target Net | 0.13 | $\sim 7422$ | Policy execution cost per control step |
| HyPoGen Weight Gen. | 3.38 | $\sim 295$ | Full hypernet run (once per task) |
| *End-to-End Control Loop (Encoder + Policy Inference)* | | | |
| *DeTaCH* (Ours) | 0.67 | $\sim 1485$ | Fastest control loop in our study |
| HyPoGen | 0.67 | $\sim 1485$ | Same target net as *DeTaCH* |
| Octo | 1.86 | $\sim 539$ | |
| Diffusion Policy (100 it denoising) | 160.17 | $\sim 6$ | Multi-step denoising at test time |

**Model-Specific Details:**

- **Octo:** 8-layer transformer, 8 attention heads, hidden dim 512

- **VQ-BeT:** Codebook size 16 with 2 groups, 8-layer transformer, 8 attention heads, hidden dim 488

- **Diffusion Policy:** 100 denoising steps, DDIM sampler, FiLM conditioning, down-scale dims [128, 256, 512], and timestep embedding dim 256

- **DiT:** 8-layer diffusion transformer, cross-attention for language and observation

- **HyPoGen:** 3 refinement blocks, hidden dim 32

- **HyperZero:** 5-layer MLP hypernetwork, hidden dim 48, ReLU activations

For all hypernet-based methods, we use the same policy network of a 5-layer MLP with hidden dim 320, resulting in a total parameter of around 0.57M.

For all methods, we keep the number of trainable parameters the same.

We compare against six state-of-the-art methods across two categories:

**Task-state Entangled Methods:**

- **Octo** (Team et al., 2024): Transformer processing concatenated language-vision tokens

- **VQ-BeT** (Shafiullah et al., 2022): Vector-quantized behavior transformer

- **Diffusion Policy** (Chi et al., 2023): UNet-based diffusion with FiLM conditioning

- **DiT** (Chi et al., 2023): Diffusion transformer with cross-attention

**Hypernetwork Methods:**

- **HyPoGen** (Ren et al., 2025): Iterative hypernetwork with optimization bias

- **HyperZero** (Rezaei-Shoshtari et al., 2023): Direct MLP mapping from language to parameters

## A.8 COMPUTATIONAL COMPARISON

*DeTaCH* follows a **"Generate Once, Act Many"** paradigm: the hypernetwork generates the policy weights once per task/language instruction, after which only the lightweight Target Net is used in the control loop. In Table 5, we show the computation cost comparison between *DeTaCH* with other baselines. *DeTaCH* and HyPoGen share the same lightweight target network, so their control-loop inference speed is effectively identical, both running comfortably in the sub-millisecond regime. Both hypernetwork-based methods are substantially faster than monolithic architectures like Octo, and orders of magnitude faster than diffusion-based policies such as Diffusion Policy. Although *DeTaCH* employs a more expressive hypernetwork and its weight generation is correspondingly slower, this cost remains real-time and is incurred only once per task. As a result, the weight-generation overhead is negligible in the overall control budget, while *DeTaCH* still enjoys the efficiency benefits of an extremely fast controller.

Table 6: Ablation of hypernetwork components on LIBERO-90. Both coarse initialization (WIN) and iterative refinement are essential for performance.

| Method Variant | Easy | Medium | Long | Overall |
|---|---|---|---|---|
| *DeTaCH* w/o WIN | 21.7 | 18.4 | 13.4 | 16.2 |
| *DeTaCH* w/ WIN only (no refinement) | **65.0** | 54.9 | 39.1 | 48.0 |
| *DeTaCH* (full) | 63.7 | **58.6** | **43.2** | **51.4** |

Table 7: Ablation on the number of refinement blocks $T$ on LIBERO-90.

| $T$ **(refinement blocks)** | **2** | **3** | **4** |
|---|---|---|---|
| **Success rate (%)** | 47.7 | **51.4** | 46.8 |

## A.9  IMPLEMENTATION DETAILS OF *DeTaCH*

**Attention-Based Implementation Details:** Our forward and backward pass simulators leverage cross-attention mechanisms to model neural gradient computation through tokenized representations. We tokenize parameter matrices $\theta_i$ of layer $i$ row-wise as $\boldsymbol{\omega}_i = \{\omega_1, \omega_2, \ldots, \omega_{n_i}\}$ using MLP encoders, while activations $z_i$ are tokenized as $\boldsymbol{\tau}_i = \{\tau_1, \tau_2, \ldots, \tau_{n_i}\}$ representing individual neurons. We use a uniform $d = 128$ for all token representations in our model. The forward simulator $F_{\text{Forward}}$ implements layer-wise computation as $\boldsymbol{\tau}_i = \text{CrossAttn}(\boldsymbol{\omega}_i, \boldsymbol{\tau}_{i-1}, \boldsymbol{\tau}_{i-1})$, where parameter tokens serve as queries attending over previous layer activations with attention head dimension $d_k = 128$. For backward pass simulation, $F_{\text{Backward}}$ estimates Jacobians using $J_{\theta_i} = \text{CrossAttn}(\boldsymbol{\tau}_{i-1}, \boldsymbol{\omega}_i, \boldsymbol{\omega}_i)$ for parameter-to-activation sensitivities and $J_{h_i} = \text{CrossAttn}(\boldsymbol{\omega}_i, \boldsymbol{\tau}_{i-1}, \boldsymbol{\tau}_{i-1})$ for inter-layer dependencies. Chain rule computations are implemented through attention-based matrix multiplications: $\frac{\partial L}{\partial z_{i-1}} = \text{CrossAttn}\left(\frac{\partial L}{\partial z_i}, J_{h_i}, J_{h_i}\right)$, ensuring modality consistency by using upstream gradients as queries while Jacobian estimates serve as both keys and values. Each cross-attention module employs 4 attention heads with 1 layer, enabling the model to capture fine-grained dependency structures essential for effective parameter generation.

### A.9.1  TRAINING DETAILS

**Data Augmentation:** We use the raw image as inputs and do not use any augmentations.

**Optimization:**

- AdamW optimizer: $lr = 0.0001$, $\beta_1$=0.9, $\beta_2$=0.999, weight decay=0.0001
- Learning rate schedule: Cosine annealing.
- Gradient clipping: Only for DiT, which uses the Max norm 1.0 to stabilize the training process.
- Mixed precision training with bf16-mixed, except for DiT, which uses fp32 since it will fail using bf16-mixed.

### A.9.2  HYPERNETWORK ARCHITECTURE ABLATION

To validate our two-stage, coarse-to-fine generation process—which combines a Weight Initialization Network (WIN) with iterative refinement—we ablate each component. The results in Table 6 show that both stages are critical. Removing the WIN module (*DeTaCH* **w/o WIN**) causes performance to collapse to 16.2%, confirming that a strong, task-conditioned initialization is essential. Conversely, using only the coarse initialization (*DeTaCH* **w/ WIN Only**) achieves a reasonable 48.0% success rate but lacks the precision for complex tasks. Our **full model**, which integrates both stages, achieves the highest performance (51.4%), notably outperforming the WIN-only variant by +4.1% on long-horizon tasks. This demonstrates that the coarse initialization and iterative refinement are complementary and crucial for the framework's success.

We also ablate the choice of number of update steps $T$ in Table 7. Increasing $T$ from 2 to 3 yields a clear improvement from 47.7% to 51.4%, confirming the benefit of a moderately deep refinement unrolling. Further increasing to $T = 4$ slightly degrades performance, likely due to over-refinement

and the added optimization difficulty, because of the introduction of WIN, *DeTaCH* is able to reach best performance with fewer update steps than HyPoGen. Based on this trade-off between performance and complexity, we set $T = 3$ for all main experiments.

### A.9.3   Ablation on Text Encoders

We further investigate how the choice of language encoder affects the quality of the generated policy parameters. Table 8 reports the success rates on LIBERO-90 when using different off-the-shelf encoders to obtain the task embedding $e_l$. *DeTaCH* achieves the highest performance when equipped with a T5-small encoder (51.4%), while replacing T5 with CLIP-base(Radford et al., 2021) or BERT-base(Devlin et al., 2019) leads to notable degradation (43.6% and 36.3%, respectively). We hypothesize that T5's text-to-text pre-training objective provides richer semantic and syntactic structure, enabling the hypernetwork to construct a more coherent parameter manifold from language input. In contrast, CLIP is primarily optimized for image–text alignment, and BERT focuses on masked-token prediction, both of which offer weaker task-level semantics for generating policy parameters. Based on these results, we adopt T5-small as our default language encoder throughout all experiments.

Table 8: Ablation on the choice of language encoder.

| Language Encoder | Success Rate (LIBERO-90) |
| --- | --- |
| T5-small | 51.4% |
| CLIP-base | 43.6% |
| BERT-base | 36.3% |

### A.9.4   Additional Results with Engineering Enhancements

In the main paper, we adopt a controlled "clean" training protocol—single third-person RGB view, no augmentation, and no action chunking or smoothing—to isolate the architectural contribution of *DeTaCH* from auxiliary engineering factors. While this ensures fair comparisons across similarly sized models (∼25M parameters), it naturally produces lower absolute success rates compared to prior works that rely heavily on multi-view inputs and extensive engineering pipelines. To address this concern, we present additional results that evaluate *DeTaCH* under increasingly strong training pipelines, beginning with full SOTA configurations.

**Comparison with BAKU and MolmoAct under Full SOTA Pipelines.**   We first evaluate *DeTaCH* under the *maximal* engineering pipeline used in BAKU and MolmoAct, incorporating multi-view RGB observations, image augmentation, temporal smoothing, and action chunking. As shown in Table 9, *DeTaCH* surpasses both BAKU and MolmoAct even when given access to the same high-capacity observation and training pipeline.

Table 9: Comparison with SOTA methods using full engineering pipelines (multi-view + augmentation + chunking).

| Method | Setting | Success Rate |
| --- | --- | --- |
| BAKU (Haldar et al., 2024) | Official (Maximal) | 90.0% |
| MolmoAct (Lee et al., 2025) | Official (Maximal) | 86.6% |
| *DeTaCH* | Maximal (Ours + Tricks) | **92.8%** |

These results confirm that *DeTaCH* can exceed state-of-the-art performance with the same engineering enhancements.

**Enhanced Baselines under the Single-View Protocol.**   Next, we apply the *same* engineering enhancements (image augmentation + action chunking) to all baselines, but keep the observation protocol constrained to a *single* third-person RGB view. This setup isolates architectural differences while still benefiting from modern training techniques.

Table 10: Enhanced baseline comparison under the single-view setting (image augmentation + action chunking).

| Method | Success Rate |
|---|---|
| Diffusion Policy | 75.23% |
| Octo | 84.36% |
| *DeTaCH* | **89.12%** |

Even with the same single-view constraints and shared enhancements, *DeTaCH* still outperforms all baselines—demonstrating that its architectural advantages persist independently of auxiliary engineering tricks.

**Fast Few-Shot Adaptation under Enhanced Training.** We also evaluate adaptation performance in the enhanced pipeline. Table 11 summarizes final success rates at 1000 gradient steps.

A notable finding is that *DeTaCH* with only **1 demonstration (55.2%)** matches or exceeds Octo with **20 demonstrations (56.0%)**.

Table 11: Final Adaptation Success Rates in LIBERO (Steps = 1000).

| Model | 1 Demo | 3 Demo | 5 Demo | 10 Demo | 20 Demo |
|---|---|---|---|---|---|
| Octo | 26.8% | 43.6% | 43.0% | 58.0% | 56.0% |
| Diffusion Policy | 0.4% | 1.0% | 0.0% | 0.0% | 0.0% |
| *DeTaCH* | **55.2%** | **54.4%** | **68.0%** | **72.0%** | **79.0%** |

Adaptation dynamics in Table 12 reveal that *DeTaCH* (1) initializes better, (2) learns substantially faster, and (3) achieves higher final performance.

Table 12: Adaptation Dynamics Across Gradient Steps.

| Setting | Model | 0 | 50 | 200 | 500 | 1000 |
|---|---|---|---|---|---|---|
| 1 Demo | Octo | 5.2 | 19.6 | 20.4 | 28.0 | 26.8 |
| | *DeTaCH* | **10.8** | **44.8** | **42.8** | **46.0** | **55.2** |
| 5 Demo | Octo | 5.2 | 18.4 | 40.0 | 43.0 | 43.0 |
| | *DeTaCH* | **10.8** | **50.4** | **63.6** | **60.2** | **68.0** |
| 20 Demo | Octo | 5.2 | 20.0 | 39.0 | 56.0 | 56.0 |
| | *DeTaCH* | **10.8** | **46.0** | **62.0** | **64.0** | **79.0** |

These results underscore the intrinsic sample efficiency of *DeTaCH*.

**(d) Enhanced Language Variation Robustness.** Finally, we revisit the language variation experiments (Table 4) using the enhanced training pipeline. As can be seen from Table 13, all models improve substantially, but *DeTaCH* remains the strongest:

*DeTaCH* therefore demonstrates robustness not only to tasks and demonstrations, but also to linguistic variation.

**Summary.** Across all evaluation settings— (1) maximal multi-view SOTA pipelines, (2) enhanced single-view baselines, (3) fast adaptation, and (4) language-variation robustness— *DeTaCH* consistently achieves the highest performance. These results confirm that *DeTaCH*'s advantages arise fundamentally from its architectural decoupling, rather than reliance on auxiliary implementation tricks.

Table 13: Enhanced language variation robustness

| Method | Diffusion Policy | Octo | *DeTaCH* |
|---|---|---|---|
| **Success Rate** | 69.3% | 86.4% | **89.4%** |

### A.9.5 REAL ROBOT EXPERIMENT

To demonstrate the real-world applicability of our method, we conducted an experiment using 100 demonstrations for 5 Pick&Place tasks. Out of these, 95 demonstrations were used for training, while the remaining 5 were held out for validation. During training, we selected the checkpoint with the smallest validation loss to evaluate the model's accuracy.

We trained *DeTaCH*, Octo, and DiffPolicy for 200 epochs on the collected data, incorporating standard data augmentations, such as random shifts, rotations, scaling, and color jittering. Additionally, all methods used an action chunk size of 16, with 2-view RGB-D images from front and left views and proprio-states as observations.

The results of these experiments are summarized in Table 14. A detailed task description mapping can be found in Table 15.

Throughout the experiment, we observed several interesting findings:

- **DiffPolicy** did not fully converge by epoch 200. In the Red Apple and Croissant tasks, the gripper frequently failed to open properly when trying to place the object. Meanwhile, in the Banana and Long Bread tasks, the policy often struggled to secure a firm grasp on the object, causing it to fall off the table midway through the task.
- **Octo** exhibited challenges, particularly with the "Banana" task, where it failed to open the gripper when trying to place the object. We suspect this issue arose due to overfitting. For the apple tasks, the gripper was overfitted to specific locations, which caused the apple to be pushed away instead of grasped. For the bread tasks, the gripper was consistently positioned higher than the expected grasp height, leading to task failure.
- *DeTaCH*, in contrast, demonstrated relatively strong performance across all tasks, including robustness in recovering from nearly failed episodes.

For visualizations of the experiment, please refer to Appendix C.

Table 14: Task Success Rates on real robot, with 10 rollouts per task, *DeTaCH* significantly outperforms baselines

| Task | Banana | Green Apple | Red Apple | Croissant | Long Bread | Success Rate |
|---|---|---|---|---|---|---|
| Diffusion Policy | 0.0 | 0.7 | 0.0 | 0.0 | 0.3 | 0.20 |
| Octo | 0.2 | 0.5 | 0.4 | 0.8 | 0.4 | 0.46 |
| *DeTaCH* | **0.7** | **0.7** | **0.9** | **0.8** | **0.7** | **0.76** |

Table 15: Task descriptions mapping on real robot

| Task Name | Task Description |
|---|---|
| Banana | Pick yellow banana and place it in basket |
| Green Apple | Pick green apple and place it in basket |
| Red Apple | Pick red apple and place it in basket |
| Croissant | Pick small croissant and place it in tray |
| Long Bread | Pick long bread and place it in tray |

### A.9.6 ANALYSIS OF LANGUAGE AND TASK DIVERSITY

In this section, we provide an analysis of how the performance of DeTaCH varies with the diversity of training instructions and tasks. We evaluate the success rates on LIBERO-90 under two different

axes of diversity: *language diversity* (number of distinct training instructions per task) and *task diversity* (number of training tasks).

**(a) Language Diversity: Number of Training Instructions**   We vary the number of distinct language instructions per task for training and test on the same extra 10 unseen instructions in LIBERO-90 (in the minimal setting without augmentation, chunking, etc.) and measure success rates. The results are shown in Table 16.

Table 16: Impact of Language Diversity (Number of Training Instructions per Task) on LIBERO-90 Success Rate.

| Num. Training Lang. | 10 | 20 | 35 | 50 |
|---|---|---|---|---|
| Octo | 24.7% | 31.2% | 24.4% | 37.2% |
| Diffusion Policy | 15.8% | 16.0% | 17.0% | 20.9% |
| DeTaCH (Ours) | **32.6%** | **31.6%** | **30.3%** | **39.8%** |

From the results, we observe that DeTaCH consistently outperforms Octo and Diffusion Policy across all diversity levels. Additionally, the performance on unseen instructions improves as the number of training instructions increases, indicating that DeTaCH's hypernetwork learns a *shared semantic structure* that is robust to varied phrasings, rather than overfitting to specific templates.

**(b) Task Diversity: Number of Tasks**   Next, we vary the number of training tasks using subsets of LIBERO-90 and report success rates in Table 17. The performance trends for each method are shown below.

Table 17: Impact of Task Diversity (Number of Tasks) on LIBERO-90 Success Rate.

| Num. Tasks | 20 | 40 | 60 | 80 |
|---|---|---|---|---|
| Octo | 43.7% | 42.6% | 41.7% | 46.1% |
| Diffusion Policy | 27.6% | 22.1% | 27.7% | 28.2% |
| DeTaCH (Ours) | **43.8%** | **49.1%** | **48.6%** | **51.4%** |

From the results, we can see that:

- Octo's performance remains relatively flat or even slightly degrades as more tasks are added. - Diffusion Policy's performance stays roughly the same as the number of tasks increases. - In contrast, DeTaCH benefits from increased task diversity, with performance improving steadily from 43.8% to 51.4%.

This scaling behavior is consistent with the view that DeTaCH's hypernetwork learns a *semantically structured parameter manifold* that becomes richer—not more confused—as more tasks are introduced.

These results highlight the advantages of DeTaCH in handling both task and language diversity, underscoring its ability to generalize more effectively than task-state entangled architectures like Octo and Diffusion Policy.

### A.9.7    DETAILED TASK DESCRIPTIONS FOR VISUALIZATION

Table 18 provides the complete mapping between the numerical indices used in Figure 3 (t-SNE embedding), the short task identifiers, and the full raw language instructions from the dataset.

### A.9.8    MANIFOLD ANALYSIS ON META-WORLD

We visualize the learned parameter manifold of Meta-World in Figure 4. The t-SNE projection of the generated policy parameters reveals a highly structured manifold where semantically distinct tasks form clear and well-separated clusters.

Table 18: Mapping of task indices to LIBERO task descriptions.

| Idx | Short ID | Full Language Instruction (Raw) |
|---|---|---|
| 1 | K3-trn-stv | KITCHEN_SCENE3_turn_on_the_stove |
| 2 | K8-trn-stv | KITCHEN_SCENE8_turn_off_the_stove |
| 3 | K6-cls-mw | KITCHEN_SCENE6_close_the_microwave |
| 4 | K7-opn-mw | KITCHEN_SCENE7_open_the_microwave |
| 5 | LR3-pck-btr-put-try | LIVING_ROOM_SCENE3_pick_up_the_butter_and_put_it_in_the_tray |
| 6 | K2-opn-tdrw-drw | KITCHEN_SCENE2_open_the_top_drawer_of_the_cabinet |
| 7 | K1-opn-bdrw-drw | KITCHEN_SCENE1_open_the_bottom_drawer_of_the_cabinet |
| 8 | K1-opn-tdrw-drw | KITCHEN_SCENE1_open_the_top_drawer_of_the_cabinet |
| 9 | K10-cls-tdrw-drw | KITCHEN_SCENE10_close_the_top_drawer_of_the_cabinet |
| 10 | K4-cls-bdrw-drw | KITCHEN_SCENE4_close_the_bottom_drawer_of_the_cabinet |
| 11 | K5-cls-tdrw-drw | KITCHEN_SCENE5_close_the_top_drawer_of_the_cabinet |
| 12 | K4-put-wine-rack | KITCHEN_SCENE4_put_the_wine_bottle_on_the_wine_rack |
| 13 | K9-put-pan-cab | KITCHEN_SCENE9_put_the_frying_pan_on_the_cabinet_shelf |
| 14 | LR6-put-pud-plt | LIVING_ROOM_SCENE6_put_the_chocolate_pudding_to_the_left_of_the_plate |
| 15 | S1-pck-bk-plc-cdy | STUDY_SCENE1_pick_up_the_book_and_place_it_in_the_front_compartment_of_the_caddy |
| 16 | LR1-pck-soup-put-bsk | LIVING_ROOM_SCENE1_pick_up_the_alphabet_soup_and_put_it_in_the_basket |
| 17 | LR2-pck-mlk-put-bsk | LIVING_ROOM_SCENE2_pick_up_the_milk_and_put_it_in_the_basket |
| 18 | S3-pck-rmug-plc-cdy | STUDY_SCENE3_pick_up_the_red_mug_and_place_it_to_the_right_of_the_caddy |
| 19 | K1-put-bkbwl-bwl | KITCHEN_SCENE1_put_the_black_bowl_on_top_of_the_cabinet |
| 20 | K10-cls-tdrw-put-bkbwl | KITCHEN_SCENE10_close_the_top_drawer_of_the_cabinet_and_put_the_black_bowl_on_top_of_it |
| 21 | K10-put-bkbwl-bwl | KITCHEN_SCENE10_put_the_black_bowl_in_the_top_drawer_of_the_cabinet |
| 22 | K5-put-bkbwl-bwl | KITCHEN_SCENE5_put_the_black_bowl_in_the_top_drawer_of_the_cabinet |
| 23 | K3-trn-stv-put-pan | KITCHEN_SCENE3_turn_on_the_stove_and_put_the_frying_pan_on_it |
| 24 | K2-stk-bkbwl-bwl | KITCHEN_SCENE2_stack_the_black_bowl_at_the_front_on_the_black_bowl_in_the_middle |
| 25 | LR5-put-rmug-mug | LIVING_ROOM_SCENE5_put_the_red_mug_on_the_left_plate |
| 26 | K1-put-bkbwl-bwl1 | KITCHEN_SCENE1_put_the_black_bowl_on_the_plate |
| 27 | LR4-stk-bwl-plc-try | LIVING_ROOM_SCENE4_stack_the_left_bowl_on_the_right_bowl_and_place_them_in_the_tray |

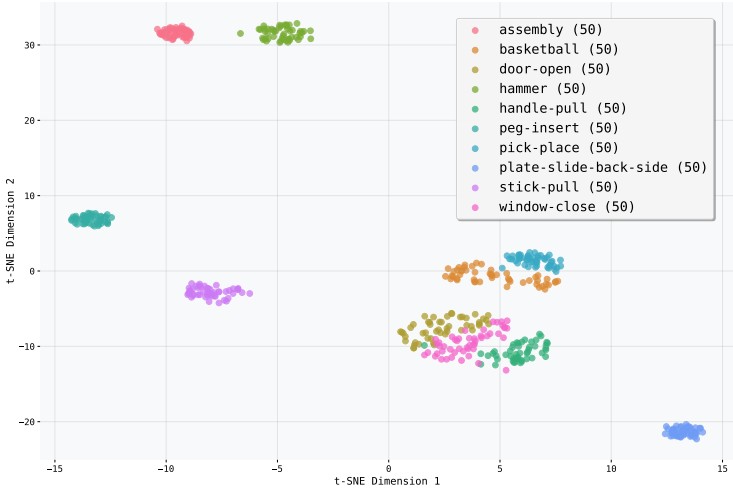

Figure 4: **t-SNE visualization of generated policy parameters for different tasks in Meta-World.** Each color represents a different task (e.g., assembly, basketball, door-open)

### A.9.9 LANGUAGE AUGMENTATION DETAILS

For robustness evaluation, we generate 50 paraphrases per task using templates:

**Original:** "put the chocolate pudding to the right of the plate"
**Augmented examples:**

- "move the chocolate pudding to the right side of the plate"
- "place the pudding on the right of the plate"
- "set the chocolate pudding beside the plate on the right"
- "shift the pudding to the plate's right side"
- "position the chocolate pudding right of the plate"

Augmentations preserve semantic meaning while varying: (1) verb choice, (2) color descriptors, (3) object synonyms, (4) grammatical structure.

## A.10 Extended Analysis on Attention and Modality Confounding

In this section, we provide detailed definitions of the visualization techniques used in Section 3.1 and further analyze the relationship between attention quality and policy performance.

### A.10.1 Visualization Methodology

To diagnose task-state entanglement, we strictly defined the attention map computation for different architectures:

- **Transformer-based Policies (e.g., Octo):** We visualize the self-attention weights between visual patches and language tokens. For each image patch $p$, we aggregate the attention weights allocated to all language tokens $T_l$ across attention heads in the relevant layers. This heatmap represents the "gaze" of the model driven by linguistic inputs.
- **Diffusion Transformers (e.g., DiT):** Following the DiT architecture, where image tokens serve as keys/values and noisy action tokens serve as queries, we visualize the cross-attention weights. This highlights which visual regions directly influence the action denoising process.

### A.10.2 The Correlation Between Attention and Performance

**Why Octo performs competitively despite imperfect attention?** A key observation is that Octo achieves reasonable success rates on simple tasks despite focusing on the gripper rather than the object. We posit this is a classic instance of *shortcut learning*. Deep networks are highly capable of fitting training distributions by correlating proprioceptive features (end-effector position) with actions, bypassing the need for semantic object grounding. However, as shown in Table 1, this shortcut strategy breaks down on **Long-Horizon** tasks (e.g., Octo: 38.3% vs. *DeTaCH*: 43.2%), where robust semantic understanding becomes necessary.

**Comparing Entanglement Severity.** Comparing visual baselines, Octo's attention maps, while imperfect, occasionally highlight relevant regions. In contrast, DiT's attention maps often collapse into non-informative regions (e.g., corners). This difference in "entanglement severity" correlates directly with our empirical results, where Octo consistently outperforms DiT.

### A.10.3 Absence of Cross-Modal Attention in *DeTaCH*

A fundamental property of *DeTaCH* is that it addresses modality confounding via architectural decoupling, not by regularizing attention maps. In our framework, language is processed solely by the hypernetwork to generate weights $\theta_\pi$. The resulting target policy takes only observations $o_t$ as input: $a_t = \pi(o_t; \theta_\pi)$. **Consequently, there are no language-to-image cross-attention maps to visualize during inference.** This is a design feature: by mathematically removing language from the control loop, we eliminate the possibility of visual gradients interfering with language representations, enforcing the decoupling by construction.

## A.11 Qualitative Analysis of task-state entangled architectures Limitations

To provide concrete evidence for our claims regarding the limitations of task-state entanglement, we conduct qualitative analyses on baseline models. These visualizations empirically demonstrate the phenomena of modality confounding and instruction signal attenuation, which motivate our shift towards an explicit decoupling architecture.

### A.11.1 Evidence of Modality Confounding via Attention Maps

To validate our hypothesis on modality confounding, we visualize the attention maps from the first and final layer of transformer-based policies, which are representative of **task-state entangled architectures**. As illustrated in Figure 5, these visualizations reveal a significant disconnect between the linguistic instruction and the model's visual focus across multiple tasks.

For instance, when instructed to "stack the black bowl..." (Row 1) or "pick up the alphabet soup" (Row 2), the Octo-style policy concentrates its attention almost exclusively on its own manipulator. Similarly, when tasked with putting the "white mug on the left plate" (Row 4), the model's

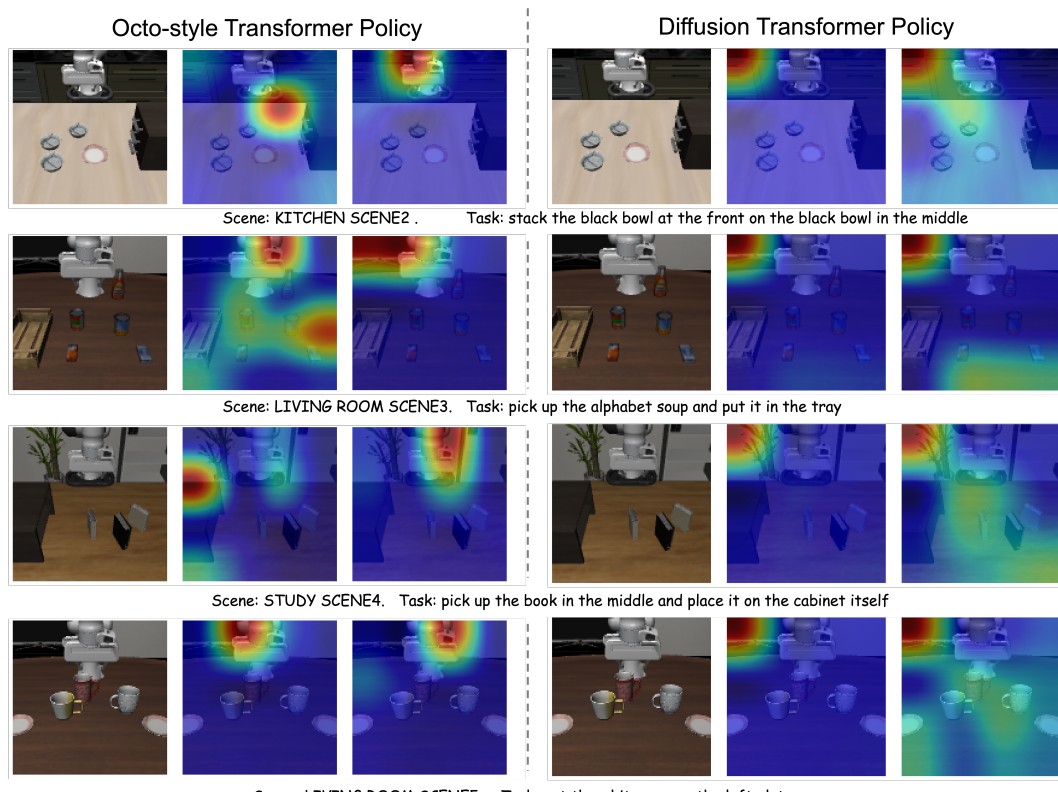

Figure 5: **Attention Map Visualization for Task-State Entangled Architectures.** Given specific language instructions (e.g., "stack the black bowl..."), the attention maps for baseline policies frequently fail to localize task-relevant objects. Instead, attention is heavily concentrated on the robot's manipulator arm or diffuse background areas, demonstrating the modality confounding problem.

focus remains fixed on its end-effector rather than the specified object. This pattern consistently demonstrates that **task-state entangled architectures** often fail to ground language instructions in the visual scene. Instead of localizing task-relevant objects, they learn to rely on spurious correlations—such as the ever-present robot arm—which is a clear manifestation of the modality confounding problem.

### A.11.2 EVIDENCE OF GRADIENT-SEMANTIC MISALIGNMENT

By analyzing the L2 norm of the gradients during backpropagation, we uncover a fundamental issue in how the policy grounds language instructions. Intuitively, for a given command, the visual features corresponding to the object of interest should receive the largest updates and thus exhibit a high gradient norm. For an instruction like "pick up the white mug," one would expect the policy's attention, as measured by gradient magnitude, to be focused on the white mug in the scene.

However, our analysis in Figure 6 reveals the opposite. The region of the image with the **highest gradient norm is consistently the robot's own manipulator**, not the object designated by the language command. This indicates that the policy's output is far more sensitive to the visual features of its own arm than to the features of the target object. The model, therefore, is not learning "what to pick up" based on the instruction, but is instead learning a generalized visual-motor pattern of its arm's movement.

This issue is further compounded by the imbalance between the gradient scales of the two modalities. The charts show that the gradient norms flowing to the visual embeddings are, on average, larger than those flowing to the language embeddings. We term this dual problem—where the visual gradient focus is misplaced and its magnitude eclipses the language signal—as **Gradient-Semantic**

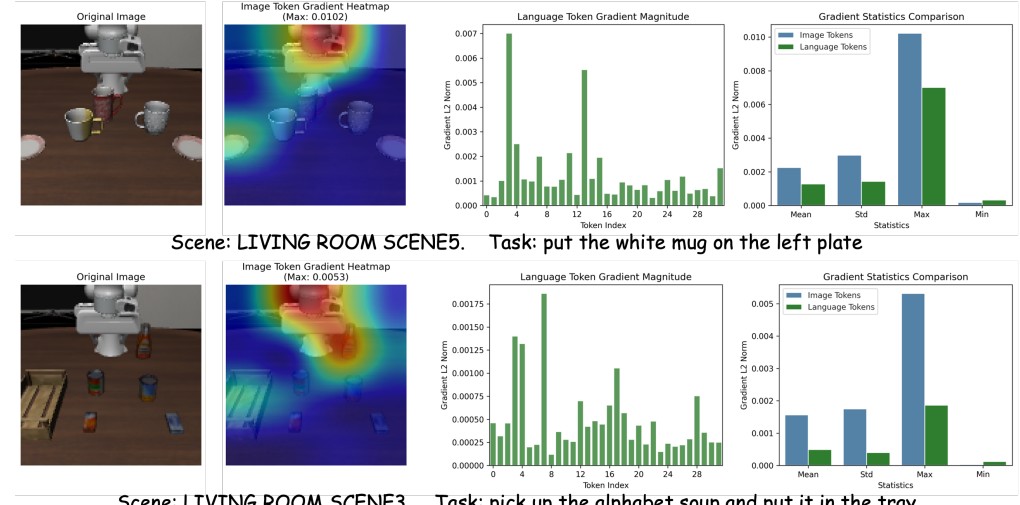

Figure 6: **Gradient Analysis for an Octo-style Policy.** Analysis of gradient norms reveals a misalignment between semantic instructions and the policy's visual focus. The heatmaps (column 2) show the highest gradient norm is on the manipulator, not the target object (e.g., "white mug"). The charts (column 4) confirm visual gradients are larger than language gradients.

**Misalignment**. This misalignment forces the model to rely on simplistic visual shortcuts rather than achieving true semantic grounding, providing a clear explanation for its brittleness on complex, language-driven tasks.

## B TASK SPLIT DETAILS

### B.1 SPLIT OF TASKS ON LIBERO

Table 19: Task categories and corresponding tasks for LIBERO

| Category | Task |
|---|---|
| One-Object Short | KITCHEN_turn_on_the_stove |
| | KITCHEN_close_the_bottom_drawer_of_the_cabinet |
| | KITCHEN_turn_off_the_stove |
| | KITCHEN_turn_on_the_stove |
| | KITCHEN_SCENE10_close_the_top_drawer_of_the_cabinet |
| | KITCHEN_open_the_top_drawer_of_the_cabinet |
| | KITCHEN_close_the_top_drawer_of_the_cabinet |
| | KITCHEN_open_the_microwave |
| | KITCHEN_open_the_top_drawer_of_the_cabinet |
| | KITCHEN_open_the_bottom_drawer_of_the_cabinet |
| | KITCHEN_close_the_microwave |
| Two-Object Short | KITCHEN3_put_the_frying_pan_on_the_stove |
| | KITCHEN3_put_the_moka_pot_on_the_stove |
| | KITCHEN4_put_the_wine_bottle_on_the_wine_rack |
| | KITCHEN4_put_the_black_bowl_on_top_of_the_cabinet |
| | KITCHEN4_put_the_black_bowl_in_the_bottom_drawer_of_the_cabinet |
| | KITCHEN6_put_the_yellow_and_white_mug_to_the_front_of_the_white_mug |
| | KITCHEN8_put_the_right_moka_pot_on_the_stove |
| | KITCHEN9_put_the_frying_pan_on_top_of_the_cabinet |
| | KITCHEN9_put_the_white_bowl_on_top_of_the_cabinet |

**Table 19 – continued from previous page**

| Category | Task |
|---|---|
| | KITCHEN9_put_the_frying_pan_on_the_cabinet_shelf |
| | KITCHEN9_put_the_frying_pan_under_the_cabinet_shelf |
| | KITCHEN10_put_the_black_bowl_in_the_top_drawer_of_the_cabinet |
| | LIVING_ROOM5_put_the_white_mug_on_the_left_plate |
| | LIVING_ROOM5_put_the_yellow_and_white_mug_on_the_right_plate |
| | LIVING_ROOM5_put_the_red_mug_on_the_left_plate |
| | LIVING_ROOM5_put_the_red_mug_on_the_right_plate |
| | LIVING_ROOM6_put_the_white_mug_on_the_plate |
| | LIVING_ROOM6_put_the_red_mug_on_the_plate |
| | LIVING_ROOM6_put_the_chocolate_pudding_to_the_left_of_the_plate |
| | LIVING_ROOM6_put_the_chocolate_pudding_to_the_right_of_the_plate |
| | KITCHEN2_put_the_black_bowl_at_the_back_on_the_plate |
| | KITCHEN2_put_the_black_bowl_at_the_front_on_the_plate |
| | KITCHEN2_put_the_middle_black_bowl_on_the_plate |
| | KITCHEN2_put_the_middle_black_bowl_on_top_of_the_cabinet |
| | KITCHEN2_stack_the_middle_black_bowl_on_the_back_black_bowl |
| | KITCHEN2_stack_the_black_bowl_at_the_front_on_the_black_bowl_in_the_middle |
| | KITCHEN5_put_the_black_bowl_in_the_top_drawer_of_the_cabinet |
| | KITCHEN5_put_the_black_bowl_on_the_plate |
| | KITCHEN5_put_the_ketchup_in_the_top_drawer_of_the_cabinet |
| | KITCHEN5_put_the_black_bowl_on_top_of_the_cabinet |
| | KITCHEN7_put_the_white_bowl_on_the_plate |
| | KITCHEN7_put_the_white_bowl_to_the_right_of_the_plate |
| | KITCHEN1_put_the_black_bowl_on_top_of_the_cabinet |
| | KITCHEN1_put_the_black_bowl_on_the_plate |
| | KITCHEN4_put_the_wine_bottle_in_the_bottom_drawer_of_the_cabinet |
| Long-Horizon | KITCHEN4_close_the_bottom_drawer_of_the_cabinet_and_open_the_top_drawer |
| | KITCHEN9_turn_on_the_stove_and_put_the_frying_pan_on_it |
| | KITCHEN10_close_the_top_drawer_of_the_cabinet_and_put_the_black_bowl_on_top_of_it |
| | KITCHEN10_put_the_butter_at_the_front_in_the_top_drawer_of_the_cabinet_and_close_it |
| | KITCHEN10_put_the_butter_at_the_back_in_the_top_drawer_of_the_cabinet_and_close_it |
| | KITCHEN10_put_the_chocolate_pudding_in_the_top_drawer_of_the_cabinet_and_close_it |
| | LIVING_ROOM1_pick_up_the_cream_cheese_box_and_put_it_in_the_basket |
| | LIVING_ROOM1_pick_up_the_alphabet_soup_and_put_it_in_the_basket |
| | LIVING_ROOM1_pick_up_the_tomato_sauce_and_put_it_in_the_basket |
| | LIVING_ROOM2_pick_up_the_orange_juice_and_put_it_in_the_basket |
| | LIVING_ROOM2_pick_up_the_milk_and_put_it_in_the_basket |
| | LIVING_ROOM2_pick_up_the_butter_and_put_it_in_the_basket |
| | LIVING_ROOM2_pick_up_the_alphabet_soup_and_put_it_in_the_basket |
| | LIVING_ROOM2_pick_up_the_tomato_sauce_and_put_it_in_the_basket |
| | LIVING_ROOM4_stack_the_left_bowl_on_the_right_bowl_and_place_them_in_the_tray |
| | LIVING_ROOM4_stack_the_right_bowl_on_the_left_bowl_and_place_them_in_the_tray |
| | LIVING_ROOM4_pick_up_the_black_bowl_on_the_left_and_put_it_in_the_tray |
| | LIVING_ROOM4_pick_up_the_salad_dressing_and_put_it_in_the_tray |
| | LIVING_ROOM4_pick_up_the_chocolate_pudding_and_put_it_in_the_tray |
| | STUDY1_pick_up_the_book_and_place_it_in_the_right_compartment_of_the_caddy |
| | STUDY1_pick_up_the_book_and_place_it_in_the_front_compartment_of_the_caddy |
| | STUDY1_pick_up_the_yellow_and_white_mug_and_place_it_to_the_right_of_the_caddy |
| | STUDY2_pick_up_the_book_and_place_it_in_the_left_compartment_of_the_caddy |
| | STUDY2_pick_up_the_book_and_place_it_in_the_right_compartment_of_the_caddy |
| | STUDY2_pick_up_the_book_and_place_it_in_the_back_compartment_of_the_caddy |
| | STUDY3_pick_up_the_book_and_place_it_in_the_left_compartment_of_the_caddy |
| | STUDY3_pick_up_the_book_and_place_it_in_the_right_compartment_of_the_caddy |
| | STUDY3_pick_up_the_book_and_place_it_in_the_front_compartment_of_the_caddy |
| | STUDY3_pick_up_the_white_mug_and_place_it_to_the_right_of_the_caddy |

**Table 19 – continued from previous page**

| Category | Task |
|---|---|
| | STUDY3_pick_up_the_red_mug_and_place_it_to_the_right_of_the_caddy |
| | STUDY4_pick_up_the_book_on_the_right_and_place_it_under_the_cabinet_shelf |
| | STUDY4_pick_up_the_book_in_the_middle_and_place_it_on_the_cabinet_shelf |
| | STUDY4_pick_up_the_book_on_the_left_and_place_it_on_top_of_the_shelf |
| | STUDY4_pick_up_the_book_on_the_right_and_place_it_on_the_cabinet_shelf |
| | LIVING_ROOM3_pick_up_the_alphabet_soup_and_put_it_in_the_tray |
| | LIVING_ROOM3_pick_up_the_tomato_sauce_and_put_it_in_the_tray |
| | LIVING_ROOM3_pick_up_the_butter_and_put_it_in_the_tray |
| | LIVING_ROOM3_pick_up_the_cream_cheese_and_put_it_in_the_tray |
| | LIVING_ROOM3_pick_up_the_ketchup_and_put_it_in_the_tray |
| | KITCHEN1_open_the_top_drawer_of_the_cabinet_and_put_the_bowl_in_it |
| | LIVING_ROOM1_pick_up_the_ketchup_and_put_it_in_the_basket |
| | STUDY1_pick_up_the_book_and_place_it_in_the_left_compartment_of_the_caddy |
| | KITCHEN3_turn_on_the_stove_and_put_the_frying_pan_on_it |

Table 20: LIBERO Eval Task File Names by Category

| Category | Task |
|---|---|
| Spatial | KITCHEN1_open_the_top_drawer_of_the_cabinet_and_put_the_bowl_in_it |
| | KITCHEN1_put_the_black_bowl_on_top_of_the_cabinet |
| | KITCHEN1_put_the_black_bowl_on_the_plate |
| | KITCHEN1_open_the_top_drawer_of_the_cabinet |
| | KITCHEN1_open_the_bottom_drawer_of_the_cabinet |
| Goal | LIVING_ROOM1_pick_up_the_ketchup_and_put_it_in_the_basket |
| | STUDY1_pick_up_the_book_and_place_it_in_the_left_compartment_of_the_caddy |
| | KITCHEN3_turn_on_the_stove_and_put_the_frying_pan_on_it |
| | KITCHEN4_put_the_wine_bottle_in_the_bottom_drawer_of_the_cabinet |
| | KITCHEN6_close_the_microwave |

## B.2 META-WORLD EVALUATION TASK SPLITS

Table 21: Task categories and corresponding tasks for Meta-World

| Category | Task |
|---|---|
| Pick & Place | pick-place |
| | pick-place-wall |
| | pick-out-of-hole |
| | shelf-place |
| | bin-picking |
| Push, Press & Pull | push |
| | stick-push |
| | coffee-push |
| | push-wall |
| | handle-press-side |
| | handle-press |
| | button-press-wall |
| | button-press-topdown |
| | button-press-topdown-wall |
| | button-press |
| | lever-pull |

**Table 21 – continued from previous page**

| Category | Task |
|---|---|
| | stick-pull |
| | handle-pull-side |
| | handle-pull |
| | coffee-pull |
| | push-back |
| Open & Close | faucet-open |
| | faucet-close |
| | window-close |
| | window-open |
| | door-open |
| | door-close |
| | drawer-close |
| | drawer-open |
| | box-close |
| Others | sweep |
| | assembly |
| | dial-turn |
| | coffee-button |
| | basketball |
| | sweep-into |
| | disassemble |
| | hammer |
| | plate-slide |
| | plate-slide-side |
| | soccer |
| | plate-slide-back-side |
| | plate-slide-back |
| | reach |
| | reach-wall |
| | peg-insert-side |
| | peg-unplug-side |
| | hand-insert |
| | door-lock |
| | door-unlock |

## C  VISUALIZATION ON REAL ROBOTS

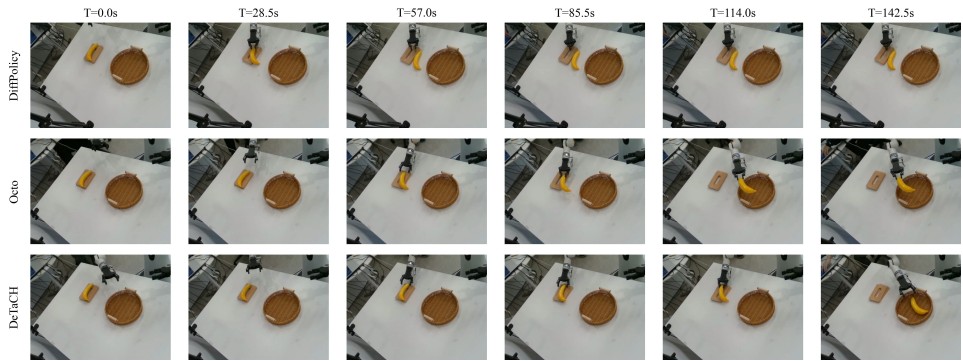

Figure 7: Visualization on task Banana

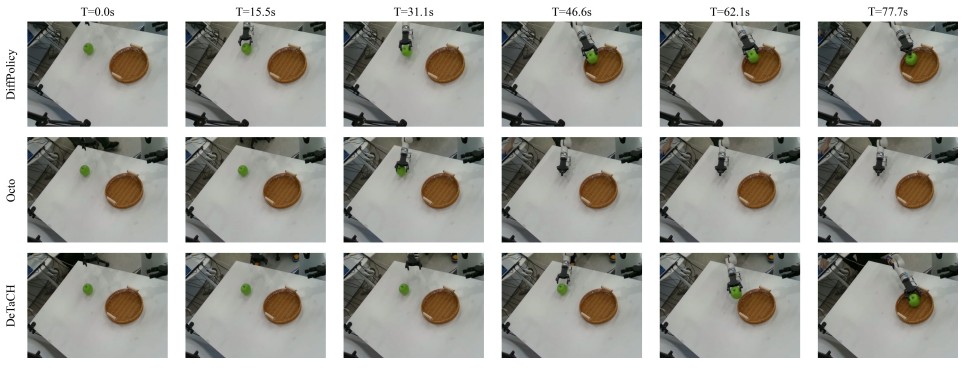

Figure 8: Visualization on task Green Apple

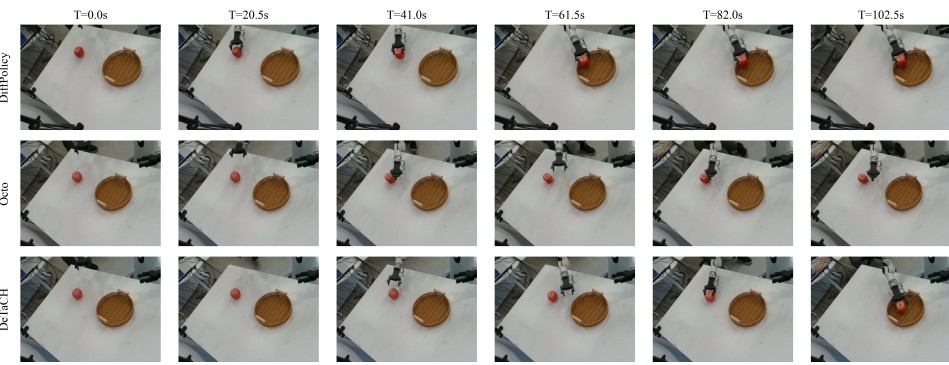

Figure 9: Visualization on task Red Apple

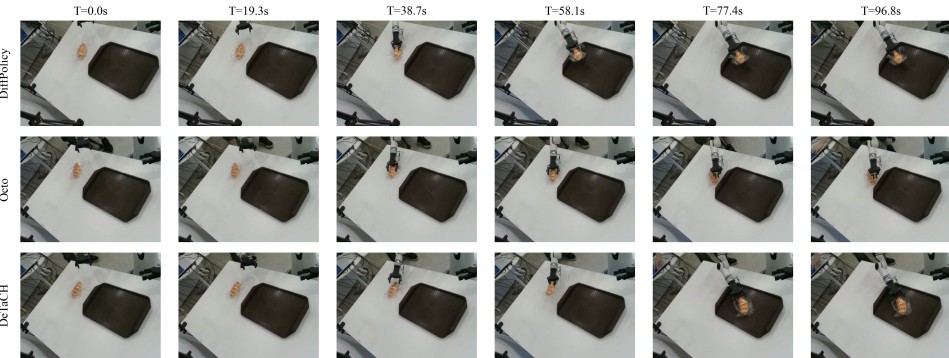

Figure 10: Visualization on task Croissant

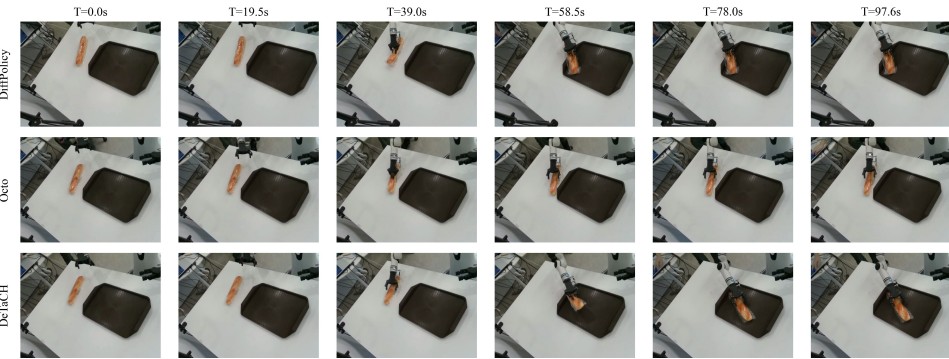

Figure 11: Visualization on task Long Bread

