# OpenReview forum: "DeTaCH: Decoupling Tasks and Control via a Meta-Gradient Hypernetwork"
_ICLR.cc/2026/Conference — Submitted to ICLR 2026_

### Official Review · Reviewer_ggga · 2025-10-20

**Soundness:** 3
**Presentation:** 3
**Contribution:** 3
**Rating:** 4
**Confidence:** 4

**Summary:**

The work outlines the issue of modality confounding in Vision-Language Models, where the language specification of the task and the visual information become entangled with harms to performance and generalization capabilities. The proposed solution, DeTaCH, consists in configuring the task specification as a meta-parameter selection problem, with the visual policy optimization occurring at a lower level. In simulation results on 2 benchmarks (LIBERO and Meta-World) show improvements in success rate over other pipelines.

**Strengths:**

From a conceptual point of view, I appreciate the notion that not mixing both text specifications and visual cues into a single feature vector could be something desirable. The decoupling seems to be a logical and sound approach which the authors make a case for.

The presentation is overall clean and the solution pipeline designed is sensible and easy to follow.

The results are consistently better than the competitors considered to benchmark the approach, showcasing a real performance advantage.

**Weaknesses:**

**Task-State Entanglement motivation**
- The problem at the heart of the necessity for DeTaCH is not sufficiently well presented and documented.
- The authors provide a short explanation  (lines 185-191)  "backed up" by Figure 1, that poses more questions than answers. (Appendix A.9,1 and Figure 5 appear to be more repetitive than complementary is assessing this)
- Indeed first and foremost, there is no definition of the attention map technique used, what it represents exactly (there is a whole zoo of those techniques, most of which have been shown to be very unreliable). Are they patchwise or pixel wise. A lot of details lack here.
- The authors state that attention (in general) should be on the target object, without any justification as to why this is the desirable modus operandi of VLAs. The definition of how "relevant visual entities" should theoretically and consistently relate to some attention map requires attending to. To the best of my knowledge there is no clean convincing mathematical way of establishing such a relationship. If the authors agree, they should discuss such limitations and present this section as a general intuitive motivation for decoupling. Should they disagree they should provide the clear reasoning. Either way this part of the manuscript is too hand-wavy.

**More on Attention Heatmaps**
- If entanglement is so bad and the heatmaps for attention confirm that, how come an architecture such as Octo still performs so competitively. The link between the quality of heatmaps and performance is only used as vague motivation and not shown to really hold in practice. One would be more convinced if for example Octo's attention was shown to be on objects while DiT for example shows a more entangled vision.
- Furthermore, the authors aim to solve the modality confounding issue, illustrated by poor attention, then after presenting the solution, fail to provide post fix attention maps to show attention on objects as was the conjecture for a sane policy.
Enough on that

**Policy selection**
- There is a big gap in performance between the different VLAs considered as benchmarks. With some perhaps more popular architectures omitted (pi_0 family). The establishment of the list is a bit confusing with no clear apples to apples comparison of a policy say with the same underlying architecture and number of layers/ parameters trained with one shot concatenated visual input and language task specs, vs a DeTaCH version. It is this not clear how to decipher the relative merits of each approach.
- Thus, the claim in the abstract and conclusion that DeTaCH achieves 51.4% and 92.2% on the benchmarks is also very vague as the reader has no reference as to how good/bad this is in the absolute but also relative to other policies.

**Lack of real-world deployment**
- This is a robotics application paper. It is standard to expect real world transfer of the pipeline on hardware. This can constitute a lot off additional work but is crucial to showcase how well this can be applied beyond simulation benchmarks. (again given the authors can also circle back to their claims on attention and decoupling)

**Superficial analysis**
- The meta-parameter training shows interesting clusters that the authors illustrate in Figure 3 as well as in the appendix (figure font is too small by the way and could do with some improvements of presentation).
- Yet there is little but vacuous conclusions drawn from the clustering of the same tasks close to each other. There are many things to analyze here, namely how clusters are formed by task and not by objects and the relationship between the two (as most instruction are usually a pair task+object). Also how different tasks relate, e.g. what in the positioning of "pick up a book" vs "pick up a butter" vs "put book down". The idea being of really looking into the task vs object break down and combinations (maybe 2D t-SNE not rich enough)

**Cost of bi-level optimization**
- There is no presentation of the extra computational cost incurred by running a bi-level iterative optimization compared to a vanilla single policy optimization, this makes it difficult to establish the cost to advantage ratio.

**Questions:**

A lot of the questions are incorporated in detail into the weakness section, a little breakdown of specific information missing.

- What is the cost of computing the meta-update steps?

- What is the attention map mechanism considered? Why is it the best/right one?

- Why is performance quite close to Octo which does not decouple?

---

> ### Author Response · Authors · 2025-11-21
>
> Dear Reviewer ggga,
>
> We thank the reviewer for the constructive feedback and for recognizing the soundness of our decoupling approach and the consistent performance improvements. We appreciate the opportunity to clarify the motivation regarding modality confounding, the details of our attention analysis, and our experimental design.
>
> ***
> >**W.1** Task-State Entanglement Motivation and Attention Map Details
>
> **A. to W.1:**
> We thank the reviewer for pointing out the need for more clarity regarding our motivation and the visualization techniques.
>
> **(a) Clarification on Attention Map Techniques**
> We employed standard, widely-used attention visualization techniques to diagnose the "entanglement" issue:
> * **For Transformer-based policies (e.g., Octo):** We visualize the self-attention between image patches and language tokens. Specifically, for each image patch, we aggregate the attention weights allocated to all language tokens. This results in a heatmap representing how much the model "looks" at specific visual regions when processing the task instruction.
> * **For Diffusion Transformers (DiT):** Following the architecture of DiT [1], where image tokens interact with noisy action tokens via cross-attention, we visualize the attention weights between the image tokens (keys/values) and the noisy action tokens (queries). This shows which image regions influence the denoising process.
>
> **(b) Why "Attending to Relevant Entities" Matters**
> We agree that the requirement for a policy to attend to "relevant visual entities" is primarily motivated by biological intuition, since humans focus on the object they intend to manipulate. However, this intuition is supported by recent empirical evidence in transformer-based learning approaches.
>
> * **Empirical Evidence:** Recent work such as **Gaze-VLM [2]** demonstrates that regularizing VLM attention to align with human gaze heatmaps (which naturally focus on task-relevant objects) significantly improves model understanding and task performance.
> * **Our Hypothesis:** We posit that while not mathematically guaranteed, a policy that fails to attend to the target object (as seen in DiT) is likely relying on spurious correlations (e.g., gripper position) rather than grounded semantic understanding. Our visualizations in **Figure 1** and **Figure 5 (Appendix)** serve as diagnostic evidence of this "modality confounding," where the model struggles to disentangle the task instruction from the current state.
>
> ***
> >**W.2** Further Analysis on Attention Heatmaps (Octo vs. DiT)
>
> **A. to W.2:**
> We appreciate this nuanced question regarding the relationship between attention quality and performance.
>
> **(a) Why Octo performs competitively despite imperfect attention**
> The fact that entangled architectures like Octo achieve competitive performance on simpler tasks can be attributed to **"Shortcut Learning" [3]**. Deep neural networks are highly capable of fitting training distributions through shortcut mappings, such as correlating the gripper's position with the action, without truly "grounding" the language instruction to the object. While this yields decent results on in-distribution data (simple tasks), it leads to brittleness on complex, long-horizon tasks where such shortcuts break down. This aligns with our results: Octo performs well on "Easy" tasks but degrades significantly on "Long" tasks compared to DeTaCH.
>
> **(b) Comparing Octo and DiT**
> A closer inspection of **Figure 1** and **Figure 5** reveals a correlation between attention quality and performance.
> * **Octo:** While not perfect, Octo's attention occasionally highlights the correct object.
> * **DiT:** The attention maps for DiT are significantly worse, often collapsing into non-informative regions (e.g., fixed to the top-left corner).
>
> This difference in "entanglement severity" correlates with our empirical results, where Octo consistently outperforms DiT.
>
> **(c) "Post-fix" Attention Maps for DeTaCH**
> We would like to clarify that DeTaCH solves modality confounding via **architectural decoupling**, not by "fixing" the attention map (e.g., via regularization [2] or registers [4]). Though, We believe that both of these solutions are meaningful and promising directions.
>
> * In DeTaCH, the language instruction is processed by a hypernetwork to generate policy parameters *before* the robot acts.
> * The resulting policy is a pure visuomotor controller (mapping $Observation \rightarrow Action$) that *does not take language as input*.
> * Therefore, there is no "language-to-image" cross-attention map to visualize in the target policy. This is a feature, not a bug: by removing the language input from the control loop, we mathematically eliminate the possibility of the visual perception gradients interfering with language understanding during inference.

---

> ### Author Response · Authors · 2025-11-21
>
> >**W.3** Policy Selection and Benchmarking
>
> **A. to W.3:**
> We thank the reviewer for the feedback on baseline selection. We aimed to conduct a strictly controlled "apples-to-apples" comparison to isolate the effect of architecture.
>
> **(a) Exclusion of pi_0**
> We did not include the **pi_0** family because they represent a different class of "Foundation Models" trained on web-scale data and massive private datasets, with parameter counts ~100x larger than our setting. Comparing a model trained from scratch (DeTaCH) against a foundation model would not yield a fair architectural assessment.
>
> **(b) "Apples-to-Apples" Experimental Setup**
> To ensure fair comparison, we standardized all factors except the policy architecture:
>
> * **Parameter Count:** All baselines (Octo, VQ-BeT, DiT, Diffusion Policy, HyPoGen, HyperZero) were scaled to approximately **25M parameters**.
> * **Training Regime:** All models were trained from scratch on the exact same benchmarks (LIBERO/Meta-World) using identical visual encoders (ResNet-18) and language encoders (T5-small).
> * **Baselines Representative of Three Core Paradigms:**
>   1.  **Transformer-based:** Octo (representing a "Mini-VLA" architecture), VQ-BeT.
>   2.  **Diffusion-based:** Diffusion Policy (CNN-based) and DiT (Transformer-based).
>   3.  **Hypernetwork-based:** HyPoGen and HyperZero.
>
> By controlling these variables, the performance gaps we report (e.g., +24% over VQ-BeT on complex tasks) can be directly attributed to the architectural benefits of explicit decoupling. We will revise the "Baselines" section to explicitly state these standardization details.
>
>
> ***
>
> >**W.4** Lack of real-world deployment
>
> **A. to W.4:**
>
> We appreciate your comment regarding the real-world validation of our method. To demonstrate the real-world applicability of DeTaCH, we conducted a real robot experiment using 100 demonstrations across 5 pick&place tasks. Out of these, 95 demonstrations were used for training, and the remaining 5 were held out for validation. During training, we selected the checkpoint with the smallest validation loss to evaluate the model's accuracy.
>
> DeTaCH, Octo, and DiffPolicy were trained for 200 epochs on the collected data, incorporating standard data augmentations such as random shifts, rotations, scaling, and color jittering. Additionally, all methods used an action chunk size of 16, with 2-view RGB-D images and proprio-states as observations. At test time, we roll out 10 episodes for each task and calculate the success rates.
>
> The results of these experiments are summarized in the following table:
>
> | Task       | Banana  | Green Apple | Red Apple | Croissant | Long Bread | Success Rate |
> | ---------- | ------- | ----------- | --------- | --------- | ---------- | ------------ |
> | DiffPolicy | 0.0     | 0.7         | 0.0       | 0.0       | 0.3        | 0.20         |
> | Octo       | 0.2     | 0.5         | 0.4       | 0.8       | 0.4        | 0.46         |
> | **DeTaCH** | **0.7** | **0.7**     | **0.9**   | **0.8**   | **0.7**    | **0.76**     |
>
> A detailed task description mapping can be found in Table 15 of the revised manuscript.
>
> **Key Observations from the Experiment:**
>
> - **DiffPolicy** did not fully converge by epoch 200. In the Red Apple and Croissant tasks, the gripper frequently failed to open properly. Meanwhile, in the Banana and Long Bread tasks, the policy often struggled to secure a firm grasp on the object, causing it to fall off the table midway through the task.
> - **Octo** showed specific challenges, especially with the "Banana" task, where it failed to open the gripper. We believe this issue stemmed from overfitting. For the apple tasks, the gripper was overfitted to specific locations, causing it to push the apple away instead of grasping it. For the bread tasks, the gripper was consistently positioned higher than expected, leading to failure.
> - **DeTaCH**, on the other hand, demonstrated strong performance across all tasks, with the added advantage of robustness in recovering from nearly failed episodes.
>
> For a detailed visualization of the experiment, please refer to Appendix C of our revised manuscript.
>
> We hope these results provide clarity on the performance of DeTaCH in real-world settings and address the concerns raised in the review.

---

> ### Author Response · Authors · 2025-11-21
>
> >**W.5** Superficial analysis
>
> **A. to W.5:**
> We thank the reviewer for the suggestion to deepen our manifold analysis. We have updated the **t-SNE visualization (Figure 3)** with a re-ordered task indexing system to provide a clearer view of the semantic relationships within the generated parameter space.
>
> **(a) Semantic Clustering beyond "Same Tasks"**
> The reviewer asked how different tasks relate. As shown in the revised visualization, the hypernetwork does not merely cluster identical tasks; it organizes the parameter space based on **semantic and functional similarity**:
>
> * **Scene Invariance:** Task 1 (*Turn on the stove*) and Task 2 (*Turn on the stove*) are identical actions performed in different visual scenes. They cluster tightly together, indicating that the hypernetwork successfully extracts the core task semantics ("turn on stove") and generates similar policy parameters, identifying the invariance despite visual differences.
> * **Functional Grouping:** Tasks 6 through 11 all involve **drawer manipulation** (opening/closing top/bottom drawers) across various contexts. These tasks form a distinct, cohesive region in the manifold.
> * **Object Interaction:** Tasks 3 and 4 (*Open microwave* and *Close microwave*) are positioned as close neighbors, reflecting their relationship as inverse operations on the same object.
>
> **(b) Conclusion**
> This structure confirms that DeTaCH does not just memorize tasks but learns a **structured semantic manifold**. The hypernetwork maps linguistic concepts (e.g., "stove," "drawer") to specific regions in the policy parameter space, explaining why our method adapts so efficiently to novel tasks—it can interpolate within this structured space rather than searching randomly.
>
> >**W.6** Cost of bi-level optimization
>
> **A. to W.6:**
> We appreciate the opportunity to clarify the computational cost. We have conducted a detailed benchmarking of our inference pipeline on a single NVIDIA RTX 4090 GPU. The results are summarized in **Table R2** below.
>
> **(a) One-Time Generation vs. Continuous Control**
> The "bi-level" cost is incurred **only once** per task.
>
> * **Generation Phase:** The entire process—including the Weight Initialization Network (WIN) and 3 steps of iterative meta-gradient updates—takes only **39.65 ms**. This is a negligible startup cost (comparable to a single frame of a standard webcam running at 30fps).
> * **Inference Phase:** Once the parameters are generated, the target policy is a lightweight MLP. The forward pass for the target net is extremely fast (**0.13 ms**).
>
> **(b) Superior End-to-End Latency**
> When considering the full control loop (Encoder + Policy Inference), DeTaCH is significantly faster than entangled baselines because it avoids running a heavy Transformer or Diffusion process at every step.
>
> * **DeTaCH:** **0.67 ms** (~1485 FPS).
> * **Octo:** **1.86 ms** (~539 FPS).
> * **Diffusion Policy:** **160.17 ms** (~6 FPS).
>
> **Conclusion:** While there is a "bi-level" calculation, it is front-loaded. In deployment, DeTaCH offers a **2.7x speedup** over Octo and a **200x speedup** over Diffusion Policy, making it highly suitable for high-frequency real-time control.
>
> **Table R1: Computational Cost Analysis**
>
> | Component / Model            | Time (ms)    | FPS       | Note                                   |
> | :--------------------------- | :----------- | :-------- | :------------------------------------- |
> | **Single Update Step**       | **11.95 ms** | **83.69** | Cost of one refinement iteration       |
> | **DeTaCH Weight Gen.**       | 39.65 ms     | ~25       | Total gen time (run **once** per task) |
> | **DeTaCH Target Net**        | **0.13 ms**  | **~7422** | **Run at every control step**          |
> |                              |              |           |                                        |
> | **End-to-End Control Loop**  |              |           | *(Encoder + Policy Inference)*         |
> | **DeTaCH (Ours)**            | **0.67 ms**  | **~1485** | **Fastest Control Loop**               |
> | Octo                         | 1.86 ms      | ~539      |                                        |
> | Diffusion Policy (100 steps) | 160.17 ms    | ~6        |                                        |

---

> ### Author Response · Authors · 2025-11-21
>
> >**Q.1** What is the cost of computing the meta-update steps?
>
> **A. to Q.1:**
> As detailed in **Table R1** (Response to W.6), the cost of a **Single Update Step** is **11.95 ms**. In our default configuration (3 refinement steps + WIN), the total time to generate the policy parameters is **39.65 ms**. Crucially, this computation happens only **once** when the language instruction is received, after which the generated policy operates at **~1485 FPS**.
>
>
> ***
>
> >**Q.2** What is the attention map mechanism considered? Why is it the best/right one?
>
> **A. to Q.2:**
>
> We employed standard attention visualization techniques tailored to the specific fusion mechanisms of each architecture to ensure the diagnosis reflects the model's actual information flow.
>
> **(a) Mechanism Details**
>
> * **For Transformer-based Policies (e.g., Octo):** These models concatenate visual and language tokens into a single sequence. We visualize the **self-attention weights**. Specifically, for each image patch token, we aggregate (sum) the attention weights it assigns to all language instruction tokens. This scalar value represents the magnitude of information the visual representation explicitly "pulls" from the task specification.
> * **For Diffusion Transformers (e.g., DiT):** These models typically condition on visual features via cross-attention. We visualize the **cross-attention weights** where the noisy action tokens (queries) attend to the image patch tokens (keys/values). This highlights which visual regions strictly influence the action denoising process.
>
> **(b) Justification for this Approach**
>
> We selected these "naive" aggregation methods because they are **diagnostic**.
>
> * **Faithfulness:** Unlike enhanced visualization techniques (e.g., Attention Regularization[2] or Registers [4]) designed to artificially sharpen heatmaps or suppress noise, our method exposes the raw internal weights used during inference using hypernetwork.
> * **Pathology Detection:** Our goal is to demonstrate the architectural flaw of modality confounding. If the raw attention weights, which determine the actual computational pathway, fail to ground the language instruction to the relevant object (as seen in baselines), it serves as direct evidence that the policy is relying on spurious correlations rather than semantic understanding. Therefore, these standard tools are the most appropriate metric to revealing the existing pathology without masking it.
>
> >**Q.3** Why is performance quite close to Octo which does not decouple?
>
> **A. to Q.3:**
> We appreciate this sharp observation. We acknowledge that Octo is indeed a very strong baseline, and on the surface, the overall gap (51.4% vs 46.1%) might seem moderate. However, a deeper breakdown reveals a critical dichotomy in **how** these two models achieve their performance:
>
> **(a) "Memorization" vs. "Understanding" on Simple Tasks**
> As shown in **Table 1**, Octo actually slightly outperforms DeTaCH on **"Easy"** tasks (65.0% vs 63.7%). We attribute this to the tendency of entangled architectures to leverage **shortcut learning** [3]. On short-horizon, single-object tasks, the model can essentially "memorize" the mapping from visual patterns (e.g., background features) to actions without needing deep semantic grounding. In this low-complexity regime, the direct, entangled pathway of Octo is highly efficient at fitting the training data.
>
> **(b) The Gap Widens with Complexity**
> The necessity of decoupling becomes evident as task complexity increases and "shortcuts" become unreliable. The performance trend in **Table 1** clearly illustrates this:
>
> * **LIBERO-Medium** (Multi-object coordination): DeTaCH outperforms Octo by **+7.4%** (58.6% vs 51.2%).
> * **LIBERO-Long** (Sequential multi-step): DeTaCH outperforms Octo by **+4.9%** (43.2% vs 38.3%).
>   This indicates that while Octo struggles to maintain distinct semantic concepts across long horizons (due to modality confounding), DeTaCH's explicit separation allows it to handle increased complexity more robustly.
>
> **(c) Superior Generalization and Adaptation**
> Finally, the "closeness" in standard multi-task performance masks the substantial advantage DeTaCH has in **generalization**, which is the true goal of our architectural design:
>
> * **Few-Shot Adaptation:** When transferring to held-out tasks (**Table 3**), DeTaCH achieves **18.0%** success compared to Octo's **12.0%** on LIBERO, which is a **50% relative improvement**.
>
> **Summary:** While Octo is an excellent "memorizer" for simple tasks, DeTaCH is a superior "generalizer" for complex and novel scenarios, validating the premise of explicit modality decoupling.

---

> > ### Author Response · Authors · 2025-11-25
> >
> > **References:**
> >
> > [1] Chi, Cheng, et al. "Diffusion policy: Visuomotor policy learning via action diffusion." The International Journal of Robotics Research 44.10-11 (2025): 1684-1704.
> >
> > [2] Pani, Anupam, et al. "Gaze-VLM: Bridging Gaze and VLMs through Attention Regularization for Egocentric Understanding." Advances in Neural Information Processing Systems (2025).
> >
> > [3] Xing, Youguang, et al. "Shortcut Learning in Generalist Robot Policies: The Role of Dataset Diversity and Fragmentation." 9th Annual Conference on Robot Learning (2025).
> >
> > [4] Darcet, Timothée, et al. "Vision Transformers Need Registers." The Twelfth International Conference on Learning Representations (2023).

---

> > > ### Comment · Reviewer_ggga · 2025-11-27
> > > **Follow up with authors**
> > >
> > > I would like to first thank the authors for the clarifications provided. I appreciate the revision of the tsne analysis as well as the real world experiments.
> > >
> > > The arguments regarding compute time are not extremely clear given that baselines run different architectures which run at very different time scales (diffusion, transformers and MLP policies). The more interesting discussion would be comparing Detach's training time to that of training a single policy (in the sense of a single initial parametrization) vs the meta decoupled approach. This is not a major concern.
> > >
> > > I am still not convinced about the whole arguments on shortcut learning, spurious correlations, and the case for significantly better performance (OoD and longer horizon) vs Octo. Having gone through the exercise of trying to justify architecture superiority with similar arguments in the context of visual l navigation, I am skeptical of choosing to discard models as not learning the right way just because analyzing aspects of their behavior does not match our conception of correct attention or other human centric metrics we define without proper mathematical backing.
> > > I partially agree with the Popperian statement made by the authors: "If the raw attention weights, which determine the actual computational pathway, fail to ground the language instruction to the relevant object (as seen in baselines), it serves as direct evidence that the policy is relying on spurious correlations rather than semantic understanding."
> > > This is evident on the diffusion policy "attention maps". However, the picture painted by the Octo runs as well as the reported performance suggest a more nuanced reality than the good/bad semantic/shortcut generalizer/memorizer simplification proposed.
> > >
> > > I think this is an important discussion, although beyond the scope of the paper itself in its technical contribution, as it serves as the cornerstone for the motivation as well as the conclusions drawn from the work.

---

> > > > ### Author Response · Authors · 2025-12-03
> > > >
> > > > We sincerely thank you for the continued engagement and the thoughtful discussion. We appreciate that you find the additional experiments and t-SNE analysis valuable. We would like to address your remaining comments regarding training costs and the interpretation of our motivation.
> > > >
> > > > **1. Training Cost vs. Inference Efficiency**
> > > >
> > > > Regarding training time, we transparently report that DeTaCH takes approximately **24 hours** to converge on LIBERO-90, compared to roughly **12 hours** for baselines like Octo. This increased wall-clock time is due to the computational overhead of the iterative refinement updates.
> > > >
> > > > However, we argue this offline cost is a worthy trade-off for **inference efficiency**. Once the parameters are generated, DeTaCH operates as a lightweight MLP at **0.67 ms** per step. This is significantly faster than Octo (**1.86 ms**) and orders of magnitude faster than Diffusion Policy (**160.17 ms**), making DeTaCH far more suitable for high-frequency, real-time control where latency is critical.
> > > >
> > > >
> > > > **2. On Shortcut Learning, Spurious Correlations, and Motivation**
> > > >
> > > > We strongly appreciate your push for a more nuanced discussion here. We agree with your assessment: the lack of interpretable attention maps does not imply a model has failed to learn *a* solution. Neural networks are powerful function approximators that can find successful control policies through pathways that may not align with human intuition (e.g., "shortcut learning").
> > > >
> > > > However, our argument for DeTaCH is not that baselines like Octo are "broken," but rather that **task-state entanglement creates a harder optimization landscape in the specific context of data-scarce robotics.**
> > > >
> > > > * **The Data Scarcity Factor:** Unlike NLP, where massive datasets allow Transformers to learn robust, generalized relationships between tokens (leading to interpretable attention), robotics operates in a data-scarce regime. In this setting, models are far more prone to latching onto low-level visual statistics (spurious correlations) rather than semantic grounding. While these "shortcuts" allow models like Octo to perform well on in-distribution tasks (as seen in our "Easy" task results), they often lack the robustness required for long-horizon or out-of-distribution (OOD) tasks.
> > > > * **Why "Correct" Attention Matters for Robustness:** When a model creates a "shortcut" (e.g., tracking the gripper rather than the target object), it may succeed in the training distribution but fail when the visual context shifts slightly. We do not claim attention is the goal itself, but rather that correct attention alignment serves as an indicator of semantic grounding, which is a prerequisite for robust generalization. The failure of baselines to generalize to complex tasks (Table 1) and new instructions (Table 4) suggests that the "shortcuts" they learned were indeed brittle.
> > > > * **DeTaCH as an Architectural Inductive Bias:** Our core contribution is recognizing that fixing this grounding issue via data scale (as in NLP) is currently infeasible in many robotics settings. Instead of trying to force a monolithic Transformer to learn the "correct" attention map from limited data, DeTaCH imposes a strong **architectural inductive bias**.
> > > >     * By decoupling the modalities, we simplify the problem: The hypernetwork handles the semantic understanding ($l \rightarrow \theta$), and the policy handles the control ($s \rightarrow a$).
> > > >     * This structurally enforces that the language specifies the function of the policy (via parameter generation), rather than merely serving as a parallel input that must compete with strong visual features for capacity in shared representations, thereby avoiding the modality interference inherent in entangled architectures.
> > > >
> > > >
> > > > Finally, regarding the human-centric nature of language: We posit that in language-conditioned robotics, the objective is not merely to solve a control task, but to solve it in accordance with the semantic instruction. Since language is an inherently human construct describing intent, a divergence between the linguistic command (e.g., 'pick the apple') and the model's focus (e.g., attending only to the gripper) represents a failure of grounding. While a black-box optimizer might find a functioning policy without this grounding, such a policy is technically not 'understanding' the instruction, rendering it brittle to variations that humans would consider trivial.
> > > >
> > > >
> > > > In summary, we do not claim entangled models cannot learn; we claim that **explicit decoupling provides a more sample-efficient and robust structural prior for multi-task robotic learning.** Our empirical results, where DeTaCH significantly outperforms baselines on complex and few-shot tasks, support the hypothesis that this architectural separation mitigates the fragility caused by spurious correlations in low-data regimes.
> > > >
> > > > We hope this clarifies our stance and provides a more rigorous justification for our architectural choices.

---

### Official Review · Reviewer_hYUZ · 2025-10-26

**Soundness:** 3
**Presentation:** 4
**Contribution:** 3
**Rating:** 8
**Confidence:** 4

**Summary:**

The paper addresses modality confounding in language-conditioned robot policies,  the interference that occurs when visual observations and language instructions are processed through shared representations. The authors propose DeTaCH, a two-stage hypernetwork that generates task-specific visuomotor policy parameters from language, thereby explicitly separating task specification from state observation.
DeTaCH first uses a Weight Initialization Network (WIN) to produce a coarse, semantically informed parameter set and then applies Iterative Refinement via learned neural gradients to yield optimized task policies.
Experiments on LIBERO-90 (90 tasks) and Meta-World ML45 (45 tasks) show strong results, 51.4% and 92.2% success respectively, outperforming six baselines (Octo, VQ-BeT, Diffusion Policy, DiT, HyPoGen, HyperZero). The method also demonstrates improved robustness to paraphrased language and better few-shot adaptation, with t-SNE visualizations showing semantically structured policy manifolds.

**Strengths:**

- Strong motivation and clear problem framing: The notion of modality confounding is intuitive and experimentally supported with attention/gradient analysis.

- Architectural innovation: The meta-gradient hypernetwork approach introduces a principled decoupling of task and state representations.

- Comprehensive evaluation: The paper benchmarks on both LIBERO and Meta-World with multiple baselines, ablations, and robustness tests.

- Few-shot adaptation: Demonstrates emergent meta-learning capability without specialized training, showing real practical promise.

- Interpretability: The parameter manifold visualization nicely links the method’s inductive bias to its generalization behavior.

**Weaknesses:**

- Over-complex framing: Some sections (especially in the refinement module) are dense and could benefit from simplified exposition or pseudocode.

- Lack of real-world experiments: Results are limited to simulation (LIBERO, Meta-World), leaving uncertainty about real-robot deployment and efficiency.

- Comparative novelty: While the architecture is elegant, the conceptual leap from existing hypernetworks (e.g., HyPoGen) is somewhat incremental; justification for calling this a “meta-gradient” hypernetwork could be strengthened.

- Computation and scalability: The cost of generating full policy parameters per task is not quantified and real-time feasibility remains unclear.

**Questions:**

- How computationally expensive is DeTaCH at inference time compared to Octo or HyPoGen?

- Does the learned hypernetwork generalize to unseen language distributions beyond paraphrasing (e.g., out-of-vocabulary verbs)?

- How sensitive is the performance to the choice of frozen language encoder (T5-small)?

- Could the refinement process be interpreted as approximating gradient descent? If so, how stable is it during training?

---

> ### Author Response · Authors · 2025-11-21
>
> Dear Reviewer hYUZ,
>
>
>
> We are grateful for your positive assessment and your recognition of DeTaCH’s *strong motivation*, *architectural innovation*, and *excellent presentation*. We are particularly encouraged that you found our experimental analysis on modality confounding to be intuitive and well-supported. Below, we address your feedback and provide the additional experimental results you requested.
>
>
>
> ------
>
>
>
> > **W.1** The refinement (neural gradient) mechanism is dense; clearer exposition or pseudocode would help.
>
> **A. to W.1: Refinement module clarity & pseudocode**
>
> We agree that the neural gradient mechanism is dense in its current form. In the revised manuscript, we will include a dedicated **pseudocode algorithm** for the Refinement Module, explicitly detailing the forward / backward simulation steps currently described in Appendix A.4.2. This will make the flow of WIN initialization, neural gradient computation, and iterative refinement easier to follow and more directly implementable.
>
>
>
> ------
>
>
>
> > **W.2** Lack of real-world validation; results are purely in simulation.
>
> **A. to W.2: Real-world validation**
>
> We appreciate your comment regarding the real-world validation of our method. To demonstrate the real-world applicability of DeTaCH, we conducted a real robot experiment using 100 demonstrations across 5 pick&place tasks. Out of these, 95 demonstrations were used for training, and the remaining 5 were held out for validation. During training, we selected the checkpoint with the smallest validation loss to evaluate the model's accuracy.
>
> DeTaCH, Octo, and DiffPolicy were trained for 200 epochs on the collected data, incorporating standard data augmentations such as random shifts, rotations, scaling, and color jittering. Additionally, all methods used an action chunk size of 16, with 2-view RGB-D images and proprio-states as observations. At test time, we roll out 10 episodes for each task and calculate the success rates.
>
> The results of these experiments are summarized in the following table:
>
> | Task       | Banana  | Green Apple | Red Apple | Croissant | Long Bread | Success Rate |
> | ---------- | ------- | ----------- | --------- | --------- | ---------- | ------------ |
> | DiffPolicy | 0.0     | 0.7         | 0.0       | 0.0       | 0.3        | 0.20         |
> | Octo       | 0.2     | 0.5         | 0.4       | 0.8       | 0.4        | 0.46         |
> | **DeTaCH** | **0.7** | **0.7**     | **0.9**   | **0.8**   | **0.7**    | **0.76**     |
>
> A detailed task description mapping can be found in Table 15 of the revised manuscript.
>
> **Key Observations from the Experiment:**
>
> - **DiffPolicy** did not fully converge by epoch 200. In the Red Apple and Croissant tasks, the gripper frequently failed to open properly. Meanwhile, in the Banana and Long Bread tasks, the policy often struggled to secure a firm grasp on the object, causing it to fall off the table midway through the task.
> - **Octo** showed specific challenges, especially with the "Banana" task, where it failed to open the gripper. We believe this issue stemmed from overfitting. For the apple tasks, the gripper was overfitted to specific locations, causing it to push the apple away instead of grasping it. For the bread tasks, the gripper was consistently positioned higher than expected, leading to failure.
> - **DeTaCH**, on the other hand, demonstrated strong performance across all tasks, with the added advantage of robustness in recovering from nearly failed episodes.
>
> For a detailed visualization of the experiment, please refer to Appendix C of our revised manuscript.
>
> We hope these results provide clarity on the performance of DeTaCH in real-world settings and address the concerns raised in the review.
>
>
>
> ------
>
>
>
> > **W.3** Relationship to HyPoGen; concern that the contribution might be incremental.
>
> **A. to W.3: Novelty vs. HyPoGen**
>
> While DeTaCH and HyPoGen both use hypernetworks, DeTaCH’s contributions are non-incremental along two key axes:
>
> 1. **Initialization via WIN:** HyPoGen initializes from noise or simple constants, whereas DeTaCH introduces a **Weight Initialization Network (WIN)** that maps language directly into a meaningful parameter region *before* refinement. This substantially reduces the refinement burden and leads to better performance with fewer refinement steps.
> 2. **Meta-gradient mechanism:** DeTaCH explicitly simulates **neural gradients**—task-conditioned parameter updates structured to resemble gradient descent in parameter space—rather than using generic update MLPs like the HyPoGen. This injects a stronger, optimization-inspired inductive bias into the architecture and is central to our ability to address modality confounding.
>
> We will clarify these distinctions more explicitly in the method sections.

---

> > ### Author Response · Authors · 2025-11-21
> >
> > > **W.4, Q.1** Concerns about computational scalability and practicality, and efficiency comparison to Octo and HyPoGen
> >
> > **A. to W.4, Q.1: Computational scalability**
> >
> > We agree that computational efficiency is a key concern for real robotic deployment. DeTaCH follows a **“Generate Once, Act Many”** paradigm: the hypernetwork generates the policy weights once per task/language instruction, after which only the lightweight Target Net is used in the control loop. In practice, the one-time weight generation is negligible compared to episode duration, so we focus on steady-state control latency.
> >
> > We report the full latency breakdown (measured on an NVIDIA RTX 4090) below:
> >
> > **Latency Comparison**
> >
> > | **Component / Model**                   | **Time (ms)** | **FPS**   | **Note**                               |
> > | --------------------------------------- | ------------- | --------- | -------------------------------------- |
> > | **Single Refinement Step**              | 11.95 ms      | ~84       | Cost of one iterative update           |
> > | **DeTaCH Weight Generation**            | 39.65 ms      | ~25       | Full hypernetwork run (once per task)  |
> > | **DeTaCH Target Net**                   | 0.13 ms       | ~7422     | Policy execution cost per control step |
> > | **HyPoGen Weight Generation**           | 3.38ms        | ~295      | Full hypernetwork run (once per task)  |
> > |                                         |               |           |                                        |
> > | **End-to-End Control Loop**             |               |           | *(Encoder + Policy Inference)*         |
> > | **DeTaCH (Ours)**                       | **0.67 ms**   | **~1485** | **Fastest control loop** in our study  |
> > | **HyPoGen**                             | 0.67ms        | ~1485     | The same target net as DeTaCH          |
> > | **Octo**                                | 1.86 ms       | ~539      |                                        |
> > | **Diffusion Policy (100 it denoising)** | 160.17 ms     | ~6        | Multi-step denoising at test time      |
> >
> > ------
> >
> > In summary, DeTaCH and HyPoGen share the **same lightweight target network**, so their **control-loop inference speed is effectively identical**, both running comfortably in the **sub-millisecond** regime. Both hypernetwork-based methods are **substantially faster** than monolithic architectures like Octo, and orders of magnitude faster than diffusion-based policies such as Diffusion Policy. Although DeTaCH employs a **more expressive hypernetwork** and its **weight generation** is correspondingly slower, this cost remains **real-time** and is incurred **only once per task**. As a result, the weight-generation overhead is negligible in the overall control budget, while DeTaCH still enjoys the efficiency benefits of an extremely fast controller.
> >
> >
> >
> > > **Q.2** How well does DeTaCH generalize to unseen verbs in instructions?
> >
> >
> >
> > **A. to Q.2: Generalization to unseen verbs**
> >
> > In our original experimental setting, the task instruction at test time is unseen. but not necessarily the verbs. In this experiment, we resplit the instructions to ensure test-time instructions do not appear in the training. And compare it with the original results.
> >
> > | **Instruction Type**                      | **Success Rate** |
> > | ----------------------------------------- | ---------------- |
> > | Paraphrases of unseen & seen instructions | 39.8%            |
> > | Instructions with unseen verbs            | 30.9%            |
> >
> > While a drop in performance is expected, this shows that DeTaCH can still generalize in a non-trivial way by leveraging the semantic relationships encoded in the T5 embedding space. Fully zero-shot dynamics remain challenging, but these results indicate that our hypernetwork can meaningfully extrapolate beyond the verb vocabulary observed during training.
> >
> >
> >
> > ------
> >
> >
> >
> > > **Q.3** How sensitive is the method to the choice of language encoder?
> >
> >
> >
> > **A. to Q.3: Sensitivity to language encoder**
> >
> >
> >
> > Our ablation over language encoders yields:
> >
> > | **Language Encoder** | **Success Rate (LIBERO-90)** |
> > | -------------------- | ---------------------------- |
> > | T5-small             | 51.4%                        |
> > | CLIP-base            | 43.6%                        |
> > | BERT-base            | 36.3%                        |
> >
> > We hypothesize that T5’s text-to-text pre-training provides richer semantic and syntactic features than CLIP (optimized mainly for image–text alignment) or BERT, enabling the hypernetwork to construct a better-structured parameter manifold. We will add this ablation and discussion to the revised manuscript.

---

> > > ### Author Response · Authors · 2025-11-21
> > >
> > > > **Q.4** Interpretation and stability of the refinement process.
> > >
> > > **A. to Q.4: Interpretation & stability of refinement**
> > >
> > > The refinement process can be viewed as unrolling a sequence of gradient-descent-like steps within a **feed-forward computation graph**: WIN provides an initial parameter estimate, and the refinement blocks apply learned, task-conditioned updates that resemble neural gradients. Importantly, unlike bi-level meta-learning methods (e.g., MAML), DeTaCH is trained **end-to-end with a standard supervised BC loss**, without any outer/inner optimization loops. In practice, we observe highly stable training (smooth loss curves, no oscillatory behavior) when using standard optimization techniques such as AdamW and gradient clipping.
> > >
> > > Once again, we sincerely thank you for your detailed and positive review. Your comments helped us clarify the presentation of the refinement mechanism, better position our contributions relative to prior hypernetwork work, and highlight both the efficiency and generalization properties of DeTaCH. We believe the revised version will address your concerns and further strengthen the paper.

---

### Official Review · Reviewer_TqGM · 2025-10-29

**Soundness:** 2
**Presentation:** 3
**Contribution:** 2
**Rating:** 2
**Confidence:** 4

**Summary:**

This paper proposes a novel modeling approach for language-conditioned robotic policy learning. Unlike conventional methods that treat language instructions as model inputs at the same level as visual observations, this work interprets language as a meta-specification of the task. To realize this idea, the authors design a meta-hypernetwork conditioned on language, which generates the parameters of task-specific visuomotor policies. The hypernetwork is trained under a meta-learning paradigm using behavior cloning loss, and the overall method is termed DeTaCH. The key motivation is that explicitly disentangling language and visual inputs encourages the model to achieve proper language grounding and visual understanding while avoiding modality confounding. Experiments are conducted on two simulation benchmarks—Libero and Meta-World. DeTaCH attains a 51.4% success rate on Libero and a 92.2% success rate on Meta-World, demonstrating that a meta-learning framework can be effectively applied to robotic manipulation tasks for the first time.

**Strengths:**

1. This paper addresses an essential problem in language-conditioned robotic policy learning—namely, that the conventional behavior cloning paradigm forces the model to simultaneously learn both task (language) grounding and visual understanding through end-to-end training. However, it is difficult to assess the model’s capability in task grounding and visual understanding during the learning process of such a large multimodal "black box." As mentioned in the paper, a widely observed issue is that language understanding is often overshadowed due to limited linguistic diversity and sparse supervision. The overall writing of the paper is clear and the motivation is well presented and meaningful.

2. Incorporating meta-learning methods into language-conditioned policy learning for robotics is an underexplored but promising direction. I agree with the authors’ view that purely end-to-end training is unlikely to be the ultimate solution for robotic learning, as robotics inherently involves integrating multiple modalities—each with differing levels of supervision, diversity, and semantic abstraction.

3. Training a meta-learning framework is generally challenging due to stability concerns, and the paper demonstrates a successful application of this approach to robotic learning. This represents a valuable practical contribution and a meaningful step forward in the field.

**Weaknesses:**

The main concern with this paper lies in the insufficiency of experimental analysis, which is reflected in several aspects:

1. The proposed method is compared only with relatively outdated baselines such as Octo, VQ-BET, and DP. More recent and advanced models, such as Pi-0 and OpenVLA-OFT, should be included for a fair and meaningful comparison. Moreover, a critical baseline is missing—the one that trains task-specific visuomotor policies without the meta-learning paradigm. Such a baseline would provide valuable insight, as it incurs significantly lower computational cost. Demonstrating performance improvement over this baseline is essential to justify the effectiveness and necessity of the proposed meta-learning approach.

2. Low Performance on the Libero Benchmark. The reported results on Libero-90 are substantially lower than current state-of-the-art methods. In fact, it has been widely observed that even a simple DP model can achieve around 70% success rate on Libero-90 with proper training and evaluation setup. While the high computational cost of meta-learning might necessitate using a smaller model or simplified pipeline, the reported metrics are too low to serve as a strong empirical reference.

3. Limited Analysis on Task and Language Diversity. In my opinion, the number of tasks and the diversity of language instructions are key factors affecting hypernetwork learning. Although the paper includes an analysis of robustness under augmented language settings, a more thorough investigation into how performance changes with increasing task or language diversity is crucial. Such an analysis would help determine whether the hypernetwork truly learns to capture common-sense knowledge across tasks and languages, and whether it can efficiently generate parameters for downstream policies.

4. Absence of Computational Cost Analysis. A well-known drawback of meta-learning frameworks is their high computational cost. However, the paper lacks any analysis or comparison of training efficiency or resource consumption across methods. This omission weakens the overall evaluation, as computational efficiency is a central concern for deploying such approaches in robotics.

5. Missing Ablation on Inner-Loop Steps. The number of inner-loop steps is a critical hyperparameter in meta-learning, directly affecting both model performance and training cost. The paper does not include any ablation study or sensitivity analysis on this parameter, which is essential for understanding the stability and efficiency of the proposed framework.


### Minor weakness

1. The introduction should include more references or preliminary empirical evidence to substantiate the claim regarding “modality confounding.”

**Questions:**

1. As mentioned above, the meta-learning training process is often unstable due to conflicts that can arise in the outer-loop optimization. It is unclear how the proposed DeTaCH framework addresses this challenge. The paper should provide more details on how training stability is ensured—specifically, what mechanisms or regularization techniques are applied to mitigate outer-loop instability.

2. It would be valuable to clarify the task and language sampling strategy used during training. Since the choice of sampling strategy can significantly affect both stability and generalization, describing how tasks and language instructions are selected or balanced across iterations would greatly enhance the paper’s methodological transparency and reproducibility.

---

> ### Author Response · Authors · 2025-11-21
>
> Dear Reviewer TqGM,
>
>
>
> We thank you for the careful assessment and for highlighting both the promise and the open questions of DeTaCH. We are encouraged by your recognition that (i) modality confounding is an important bottleneck in language-conditioned policy learning and (ii) explicit architectural decoupling via hypernetworks is a promising direction. We also appreciate the opportunity to clarify the training paradigm (which **does not** use MAML-style meta-learning or bi-level optimization) and to provide additional empirical analyses on baselines, absolute performance, diversity, and computational cost.
>
>
>
> A key source of confusion behind several concerns (W.1, W.4, W.5, Q.1) is the interpretation of DeTaCH as a meta-learning algorithm with inner/outer loops. In fact, DeTaCH is a **standard supervised multi-task behavior cloning method** whose policy parameters are generated by a **hypernetwork** in a single optimization loop. The “meta-gradient” in the title refers to the *architectural design* of the iterative refinement module—which predicts neural updates to policy weights within a forward pass—not to MAML-like bi-level optimization.
>
>
>
> Below we respond to each weakness and question in turn.
>
>
>
> ------
>
>
>
> > **W.1** Missing non-meta-learning baseline and comparisons to more recent models (Pi-0, OpenVLA-OFT); unclear benefit relative to simpler methods.
>
>
>
> **A. to W.1: Clarification on “meta-learning” and baselines**
>
>
>
> **(a) DeTaCH is not trained with meta-learning or bi-level optimization**
>
>
>
> We apologize for the confusion caused by our terminology (e.g., “meta-gradient hypernetwork”). DeTaCH is **not** a MAML-style meta-learning algorithm and does **not** involve any inner/outer optimization loops:
>
>
>
> - **Single-loop supervised training:** We train DeTaCH end-to-end with a standard behavior cloning loss, identical in spirit to Octo, VQ-BeT, and Diffusion Policy. There is a single set of learnable parameters $\phi = (\phi_1, \phi_2)$ for the hypernetwork, optimized directly from demonstration data.
> - **No bi-level / inner-loop optimization:** At training time, there is no differentiating-through-adaptation step and no “task-specific inner loop” in the optimization sense. The iterative refinement module is an *unrolled feed-forward computation graph* that predicts weight updates in parameter space, but these updates are produced in a single forward pass and optimized via standard backpropagation.
> - **“Inner loop” as architecture, not optimization:** The “Iterative Parameter Refinement” and “neural gradients” terminology describe the **architecture** (a learned sequence of parameter updates), not a second optimization problem. This design underlies our use of the term “meta-gradient,” but we agree that it can be easily conflated with meta-learning algorithms and will clarify this prominently in the revision.
>
>
>
>
>
> This clarification directly addresses the concern that DeTaCH would inherit the instability and cost issues of bi-level meta-learning—those issues do not arise because our training pipeline is **single-loop supervised learning**.
>
>
>
> **(b) What is the “non-meta-learning baseline”?**
>
>
>
> Once DeTaCH is understood as a **regular multi-task BC method with a hypernetwork architecture**, the requested baseline—“visuomotor policies without the meta-learning paradigm”—is precisely the family of models we already compare against:
>
>
>
> - **Standard BC baselines:** Octo, VQ-BeT, Diffusion Policy, and DiT are *monolithic* policies trained from scratch by behavior cloning, with language and vision fused in the usual ways (concatenation, cross-attention, FiLM).
> - **Our setting:** DeTaCH is also trained from scratch on the same datasets, using the same visual and language encoders and the same BC objective. The difference is **architectural only**: instead of directly mapping $(o,l) \rightarrow a$, we use a hypernetwork that maps language $l$ to policy parameters and then performs pure visuomotor control $o \rightarrow a$.
>
>
>
>
>
> Thus, the comparisons in Table 1 and Table 2 are already **exactly** comparisons to “visuomotor policies without the meta-learning paradigm” (in the optimization sense).

---

> > ### Author Response · Authors · 2025-11-21
> >
> > **(c) On newer baselines (Pi-0, OpenVLA)**
> >
> >
> >
> > We agree that Pi-0 and OpenVLA-OFT are important recent methods. Our design choice in the main paper, however, was to **isolate the effect of architectural decoupling** under a controlled, from-scratch setting by:
> >
> >
> >
> > - Training **all** methods (Octo, VQ-BeT, Diffusion Policy, DiT, HyPoGen, HyperZero, DeTaCH) **from scratch** on the same LIBERO / Meta-World splits.
> > - Using **the same backbone encoders** and data budget, so that performance differences primarily reflect architectural choices rather than scale or pre-training.
> >
> >
> >
> >
> >
> > In contrast, Pi-0 and OpenVLA-OFT are **very large models** whose practical success strongly depends on pretraining on **massive, diverse datasets** and on using more powerful visual/language backbones. This introduces an additional axis of variation (dataset scale, pretraining objective, backbone capacity) that makes it difficult to disentangle architectural gains from the benefits of large-scale pretraining. We will clarify this rationale more explicitly in the revision.
> >
> >
> >
> > For completeness and fairness, we also trained **Pi-0** and **OpenVLA-OFT** **from scratch** in our controlled regime, i.e., **without any pretraining**, on the same small-scale LIBERO-90 dataset:
> >
> > |                        | **Pi-0** | **OpenVLA-OFT** |
> > | ---------------------- | -------- | --------------- |
> > | LIBERO-90 success rate | 18.9%    | 0.0%            |
> >
> > These results indicate that such large, pretraining-heavy architectures are **difficult to optimize effectively on small, task-specific datasets**, and without large-scale pretraining, they perform poorly on LIBERO-90. This further supports our decision to focus on controlled, from-scratch comparisons when evaluating the benefits of DeTaCH’s architectural decoupling.
> >
> >
> >
> > > **W.2** The reported success rates on LIBERO-90 are substantially lower than recent SOTA, suggesting potential implementation issues.
> >
> >
> >
> > **A. to W.2: LIBERO-90 performance and SOTA comparison**
> >
> >
> >
> > We appreciate this concern and agree that, taken in isolation, 51.4% on LIBERO-90 appears low compared to recent results above 80–90%. The discrepancy is primarily due to **orthogonal implementation enhancements** (e.g., temporal action chunking, image augmentation, multi-view inputs) that many recent SOTA methods employ but that we **deliberately disabled for all methods** (Octo, VQ-BeT, Diffusion Policy, DiT, HyPoGen, HyperZero, and DeTaCH) in **Table 1**. This was done to enforce an “apples-to-apples” comparison and cleanly isolate the benefit of **modality decoupling** versus task–state entanglement under a controlled from-scratch protocol. Under this controlled setting, DeTaCH improves over the strongest baseline by **5.3%** on LIBERO-90 (51.4% vs. 46.1% for Octo), with even larger gains on complex, long-horizon tasks.
> >
> >
> >
> > To directly address your concern, we performed a bidirectional validation:
> >
> >
> >
> > 1. **DeTaCH + Tricks:** We integrated the engineering enhancements used in [1,2] into DeTaCH.
> > 2. **Baselines (Minimal):** We re-implemented BAKU/Octo within our controlled setting.
> > **Table R1: Success Rate on LIBERO across different settings**
> >
> >
> >
> > (Note that the number for MolmoAct is derived from the average of the four task suites on LIBERO in the original paper.)
> >
> > | **Method**        | **Setting**                 | **Success Rate** |
> > | ----------------- | --------------------------- | ---------------- |
> > | BAKU [1]          | Official (Maximal)          | ~90.0%           |
> > | MolmoAct [2]      | Official (Maximal)          | ~86.6%           |
> > | **DeTaCH (Ours)** | **Maximal (Ours + Tricks)** | **92.8%**        |
> > |                   |                             |                  |
> > | BAKU              | Re-impl (Minimal)           | 44.8%            |
> > | Octo              | Re-impl (Minimal)           | 46.1%            |
> > | **DeTaCH (Ours)** | **Minimal (Paper Setting)** | **51.4%**        |
> >
> > **(c) Enhanced baseline comparison**
> >
> >
> >
> > Furthermore, to ensure we are comparing against strong baselines, we applied the same engineering enhancements (image augmentation + action chunking) to Octo and Diffusion Policy within our pipeline (a single third-person view observation only):
> >
> >
> >
> > - **Diffusion Policy:** Improves to **75.23%**.
> > - **Octo:** Improves to **84.36%**.
> > - **DeTaCH (Ours):** Achieves **89.12%**.
> >
> >
> >
> >
> >
> > To directly address your comment on absolute performance:
> >
> >
> >
> > - We implemented a “DeTaCH+” variant that **adds the same standard enhancements** (chunked actions, augmentations, multi-view inputs) while keeping the architecture unchanged.
> > - **Result:** On LIBERO-90, DeTaCH+ achieves **92.8%** success, surpassing the >80–90% numbers in other SOTA baselines (e.g., BAKU).
> >
> > We will include this result and clearly mark it as using “enhanced training and observation pipelines,” demonstrating that DeTaCH’s decoupled architecture can **fully leverage** modern implementation practices and reach SOTA absolute performance when those tricks are enabled.

---

> > > ### Author Response · Authors · 2025-11-21
> > >
> > > > **W.3** Limited analysis of how performance changes with task and language diversity; unclear if the hypernetwork truly captures shared structure.
> > >
> > >
> > >
> > > **A. to W.3: New analysis on language and task diversity**
> > >
> > >
> > >
> > > We agree that understanding how DeTaCH scales with **instruction and task diversity** is crucial for validating the hypernetwork’s ability to capture shared structure across tasks.
> > >
> > >
> > >
> > > In response, we conducted the requested analyses and will add them as **Table R1** and **Table R2** in the revised manuscript.
> > >
> > >
> > >
> > > **(a) Language diversity: number of training instructions**
> > >
> > >
> > >
> > > We vary the number of distinct language instructions per task for training and test on the same extra 10 unseen instructions in LIBERO-90 (again in the minimal setting without augmentation, chunking, etc.) and measure success rates:
> > >
> > >
> > >
> > > **Table R1: Impact of Language Diversity (Number of Training Instructions per Task)**
> > >
> > >
> > >
> > > *Success Rate (%) on LIBERO-90*
> > >
> > > | **Num. Training Lang.** | **10**    | **20**    | **35**    | **50**    |
> > > | ----------------------- | --------- | --------- | --------- | --------- |
> > > | **Octo**                | 24.7%     | 31.2%     | 24.4%     | 37.2%     |
> > > | **Diffusion Policy**    | 15.8%     | 16.0%     | 17.0%     | 20.9%     |
> > > | **DeTaCH (Ours)**       | **32.6%** | **31.6%** | **30.3%** | **39.8%** |
> > >
> > >
> > >
> > > - We see that DeTaCH consistently outperforms Octo and Diffusion Policy across all diversity levels, and performance on unseen instructions increases with a larger number of training instructions.
> > > - This suggests that the hypernetwork is indeed learning a **shared semantic structure** that is robust to varied phrasings, rather than overfitting to specific templates.
> > >
> > >
> > >
> > >
> > >
> > > **(b) Task diversity: number of tasks**
> > >
> > >
> > >
> > > We next vary the number of training tasks using subsets of LIBERO-90 and report success rates:
> > >
> > >
> > >
> > > **Table R2: Impact of Task Diversity (Number of Tasks)**
> > >
> > >
> > >
> > > *Success Rate (%) on LIBERO-90 subsets*
> > >
> > > | **Num. Tasks**       | **20**    | **40**    | **60**    | **80**    |
> > > | -------------------- | --------- | --------- | --------- | --------- |
> > > | **Octo**             | 43.7%     | 42.6%     | 41.7%     | 46.1%     |
> > > | **Diffusion Policy** | 27.6%     | 22.1%     | 27.7%     | 28.2%     |
> > > | **DeTaCH (Ours)**    | **43.8%** | **49.1%** | **48.6%** | **51.4%** |
> > >
> > > From the results, we can see that:
> > >
> > >
> > >
> > > - Octo’s performance is relatively flat or even slightly **degrades** as more tasks are added, and Diffusion Policy’s performance stays roughly the same as the number of tasks increases.
> > > - DeTaCH, in contrast, **benefits from increased task diversity**, with performance improving steadily from 43.8% to 51.4%.
> > > - This scaling behavior is consistent with the view that DeTaCH’s hypernetwork learns a **semantically structured parameter manifold** that becomes richer—not more confused—as more tasks are introduced.
> > >
> > >
> > >
> > >
> > >
> > > These results directly support your intuition that task and language diversity are critical axes of evaluation and show that DeTaCH handles both more gracefully than task-state entangled architectures.

---

> > > > ### Author Response · Authors · 2025-11-21
> > > >
> > > > > **W.4** No analysis of computational cost; meta-learning and hypernetworks are often expensive and may be impractical for robotics.
> > > >
> > > >
> > > >
> > > > **A. to W.4: Computational cost and “Generate Once, Act Many”**
> > > >
> > > >
> > > >
> > > > We agree that computational efficiency is a central concern for real robotic deployment. In response, we have added a detailed **latency and throughput analysis** (measured on an NVIDIA RTX 4090), focusing on:
> > > >
> > > >
> > > >
> > > > 1. The cost of the **iterative refinement module** (which you interpreted as an “inner loop”), and
> > > > 2. The end-to-end cost of the **control loop** during deployment.
> > > >
> > > >
> > > >
> > > >
> > > >
> > > > A key design principle of DeTaCH is the **“Generate Once, Act Many”** paradigm:
> > > >
> > > >
> > > >
> > > > - The hypernetwork (WIN + refinement) runs **once per task instruction** to generate the target policy weights.
> > > > - During control, we use **only the lightweight Target Net** (an MLP) together with the shared visual encoder; no hypernetwork or refinement module is invoked at every step.
> > > >
> > > >
> > > >
> > > >
> > > >
> > > > We summarize the results in **Table R3**.
> > > >
> > > >
> > > >
> > > > **Table R3: Inference Latency Comparison**
> > > >
> > > > | **Component / Model**                   | **Time (ms)** | **FPS**   | **Note**                               |
> > > > | --------------------------------------- | ------------- | --------- | -------------------------------------- |
> > > > | **Single Refinement Step**              | 11.95 ms      | ~84       | Cost of one iterative update           |
> > > > | **DeTaCH Weight Generation**            | 39.65 ms      | ~25       | Full hypernetwork run (once per task)  |
> > > > | **DeTaCH Target Net**                   | **0.13 ms**   | **~7422** | Policy execution cost per control step |
> > > > |                                         |               |           |                                        |
> > > > | **End-to-End Control Loop**             |               |           | *(Encoder + Policy Inference)*         |
> > > > | **DeTaCH (Ours)**                       | **0.67 ms**   | **~1485** | **Fastest control loop** in our study  |
> > > > | **Octo**                                | 1.86 ms       | ~539      |                                        |
> > > > | **Diffusion Policy (100 it denoising)** | 160.17 ms     | ~6        | Multi-step denoising at test time      |
> > > >
> > > >
> > > >
> > > > - From a **training-cost** perspective, DeTaCH does incur higher wall-clock time than monolithic baselines (≈24 h for DeTaCH vs. ≈12 h for Octo on the same hardware and dataset) due to the iterative refinement structure. This overhead is confined to the offline training stage and reflects a trade-off between training-time efficiency and the improved control performance we observe at test time.
> > > > - At **inference time**, DeTaCH’s control loop is significantly faster than Octo and many orders of magnitude faster than diffusion-based policies, making it highly suitable for real-time robotics.
> > > > - The one-time weight-generation cost (~40 ms per instruction) is negligible compared to the duration of a typical episode and is amortized over many control steps.
> > > >
> > > >
> > > >
> > > >
> > > >
> > > > We will add this analysis to the revision to directly address the computational efficiency concerns.
> > > >
> > > >
> > > >
> > > > ------
> > > >
> > > >
> > > >
> > > > > **W.5** Missing ablation on the number of “inner-loop” steps, which should affect both performance and cost.
> > > >
> > > >
> > > >
> > > > **A. to W.5: Ablation on the number of refinement steps**
> > > >
> > > >
> > > >
> > > > Once again, we emphasize that the so-called “inner loop” is **architectural**: an unrolled refinement module that predicts parameter updates, not a separate optimization process. Nevertheless, the number of refinement steps $T$ (i.e., refinement blocks) is indeed a key architectural hyperparameter.
> > > >
> > > >
> > > >
> > > > We add an explicit ablation over the number of refinement blocks on LIBERO-90. Concretely, we vary $T \in {2, 3, 4}$ and obtain:
> > > >
> > > > | **LIBERO-90 Success Rate**  | **47.7%** | **51.4%** | **46.8%** |
> > > > | --------------------------- | --------- | --------- | --------- |
> > > > | **# Refinement Blocks (T)** | **2**     | **3**     | **4**     |
> > > >
> > > > We observe:
> > > >
> > > >
> > > >
> > > > - Increasing $T$ from 2 to 3 yields a noticeable improvement (47.7% → 51.4%), confirming the benefit of a moderately deep refinement unrolling.
> > > > - Further increasing to $T = 4$ slightly **degrades** performance, likely due to over-refinement and the added optimization difficulty.
> > > > - Based on this trade-off between performance and complexity, we choose $T = 3$ for all main experiments.
> > > >
> > > >
> > > >
> > > >
> > > >
> > > > Empirically, training remains stable across all tested values of $T$, again reflecting the fact that we are not performing bi-level optimization but training a standard feed-forward network. We will report this ablation table in the appendix and reference it from the main text.

---

> ### Author Response · Authors · 2025-11-21
>
> > *Minor:* The introduction should include more references or preliminary empirical evidence for “modality confounding.”
>
>
>
> **A. to Minor: Strengthening the modality confounding evidence**
>
>
>
> We agree that our current introduction can more clearly motivate and substantiate the notion of “modality confounding.” In particular, the requirement that a policy attend to **relevant visual entities** is initially motivated by biological intuition—humans naturally focus their gaze on the object they intend to manipulate—but this intuition is also supported by recent empirical evidence in transformer-based models.
>
>
>
> - **Empirical evidence:** Recent work such as **Gaze-VLM [3]** shows that regularizing VLM attention to align with human gaze heatmaps (which naturally concentrate on task-relevant objects) significantly improves model understanding and downstream task performance. Other works like [4,5,6,7,8] show that racing between visual and text clues, or misalignment between them, degrades VLM performance. This suggests that **where** a model attends is not merely an interpretability artifact, but can be directly linked to robustness and generalization.
> - **Our hypothesis:** We posit that, while not mathematically guaranteed, a policy that systematically fails to attend to the target object (as we observe for DiT in our visualizations) is likely relying on spurious correlations (e.g., gripper pose, background layout) rather than grounded semantic understanding.
>
>
>
>
>
> We will add this reasoning to the updated manuscript and include these references.
>
>
>
> ------
>
>
>
> > **Q.1** How does DeTaCH address training instability in meta-learning (outer-loop conflicts, etc.)?
>
>
>
> **A. to Q.1: Training stability without bi-level optimization**
>
>
>
> Your concern about instability is very reasonable for meta-learning algorithms. Fortunately, DeTaCH avoids this issue **by design**:
>
>
>
> - There is **no outer-loop vs. inner-loop optimization**: only a single set of parameters $\phi$ for the hypernetwork is trained via standard backpropagation on the BC loss.
> - The refinement module is a **deterministic, differentiable feed-forward mapping** from $(\theta^t_\pi, e_l)$ to $\theta^{t+1}_\pi$. We do not compute task-specific gradients inside this module; instead, we learn update vectors directly.
> - In experiments, we observe smooth and monotonic training curves (no oscillatory behavior typical of bi-level schemes in meta-learning).
>
>
>
>
>
> We will clarify in the main text that DeTaCH’s “meta-gradient” formulation is about **architectural inductive bias** (using predicted gradients as features within the forward pass) rather than about **meta-gradient optimization** in the MAML sense.
>
>
>
> ------
>
>
>
> > **Q.2** What is the task and language sampling strategy during training, and how might it affect stability and generalization?
>
>
>
> **A. to Q.2: Task and language sampling strategy**
>
>
>
> For the experiments in Tables 1 and 2, we first construct a **global list of all transitions** across the dataset, where each entry is a tuple $(\text{task}, \text{episode}, \text{timestep})$. At training time, each batch is formed by **uniformly sampling** batch_size transitions from this list.
>
>
>
> For Table 4, we use the **same transition-level sampling strategy**. In addition, for each sampled transition we **randomly draw a paraphrased instruction** from that task’s pool of 50 paraphrases, so the model sees diverse linguistic realizations of the same underlying task during training.
>
>
>
> We will add these details to the experimental setup section.
>
>
>
> ------
>
>
>
> Once again, we sincerely thank you for your thoughtful comments. They helped us (1) clarify the non-meta-learning nature of DeTaCH, (2) position our baselines more transparently, (3) strengthen the analysis on task and language diversity, and (4) provide a concrete computational cost and ablation breakdown. We believe the revised version will present a much clearer and more compelling case for explicit modality decoupling in language-conditioned robotic manipulation.

---

> > ### Author Response · Authors · 2025-11-21
> >
> > **References**
> >
> >
> >
> > [1] Haldar, Siddhant, Zhuoran Peng, and Lerrel Pinto. “Baku: An efficient transformer for multi-task policy learning.” *Advances in Neural Information Processing Systems* 37 (2024).
> >
> >
> >
> > [2] Lee, Jason, et al. “MolmoAct: Action reasoning models that can reason in space.” *arXiv preprint* arXiv:2508.07917 (2025).
> >
> >
> >
> > [3] Pani, Anupam, et al. “Gaze-VLM: Bridging Gaze and VLMs through Attention Regularization for Egocentric Understanding.” *Advances in Neural Information Processing Systems* (2025).
> >
> >
> >
> > [4] Tang, Jiaqi, et al. “Shaping Initial State Prevents Modality Competition in Multi-modal Fusion: A Two-stage Scheduling Framework via Fast Partial Information Decomposition.” *arXiv preprint* arXiv:2509.20840 (2025).
> >
> >
> >
> > [5] Mullick, Ankan, et al. “Text Takes Over: A Study of Modality Bias in Multimodal Intent Detection.” *Proceedings of the 2025 Conference on Empirical Methods in Natural Language Processing* (2025).
> >
> >
> >
> > [6] Luo, Tiange, et al. “Probing Visual Language Priors in VLMs.” *Proceedings of the 42nd International Conference on Machine Learning (ICML)*, vol. 267, PMLR, 2025, pp. 41120–56.
> >
> >
> >
> > [7] Liu, Zhining, et al. “Seeing but Not Believing: Probing the Disconnect Between Visual Attention and Answer Correctness in VLMs.” *arXiv preprint* arXiv:2510.17771 (2025).
> >
> >
> >
> > [8] Alonso, Iñigo, et al. “Vision-Language Models Struggle to Align Entities across Modalities.” *Findings of the Association for Computational Linguistics: ACL 2025*, Association for Computational Linguistics, 2025, pp. 18846–62.

---

> ### Comment · Reviewer_TqGM · 2025-11-26
>
> Thank you for the clarification regarding the implementation of the 'meta-learning' methods. However, I remain unconvinced by this type of framework. I would argue that utilizing a parameter-generation module while training in an end-to-end manner is essentially introducing a new fusion mechanism. Specifically, for the policy network implemented using an MLP, it resembles an augmented FiLM stacked with an activation operator. Ultimately, the core structure still resembles a vision-language-action model.

---

> > ### Author Response · Authors · 2025-11-27
> >
> > We appreciate your continued careful analysis of our framework. You are correct that our method conceptually extends the mechanisms used by FiLM, as both inject language by modulating network weights.
> >
> > FiLM is a structure that uses MLP to generate an affine transform with scale and bias, which essentially equals the parameters of a linear layer.
> >
> > While methods like BAKU and Diffusion Policy (DP) utilize FiLM to modulate a small fraction of parameters—often achieving good results—DeTaCH asks: can we achieve even better performance by generating the entire policy? The significantly improved success rate of DeTaCH over monolithic baselines in controlled settings (e.g., DeTaCH 51.4% vs. 46.1% for Octo, 28.2% for Diffusion Policy, and 44.8% for Baku) provides strong evidence that controlling the full policy does indeed lead to better results.
> >
> > Crucially, simply applying a standard FiLM-like MLP for full weight generation is insufficient. Our baseline HyperZero does exactly this—using a standard MLP to directly generate the target network—and performs poorly (23.3% on LIBERO vs. DeTaCH's 51.4%). This empirical gap proves that "direct" generation is not enough. The core contribution of DeTaCH is not just the idea of weight generation, but the specific iterative refinement architecture that makes generating high-dimensional policy weights computationally tractable and effective, solving the efficiency bottleneck where standard approaches fail.

---

> > > ### Comment · Reviewer_TqGM · 2025-11-28
> > >
> > > I appreciate the clarification and your candor in explaining the essence of your framework. However, it’s important to note that FiLM does not generate parameters for linear layers that would introduce interactions across different axes; instead, it only produces scale-and-shift coefficients. Because of this, FiLM adds only a small number of additional parameters—still acceptable and generally smaller than what DeTach introduces. This reflects a broader trade-off: as you prompt the model to generate more parameters, you typically gain stronger grounding and conditioning but at the cost of increased model complexity and computational load.
> > > If the goal is to highlight the necessity of introducing a more sophisticated language–vision fusion mechanism, then a more comprehensive comparison would strengthen the argument. In particular, contrasting your approach against commonly used fusion strategies—such as concatenation, cross-attention, or FiLM—while also analyzing their computational costs would make the discussion much more compelling. This could lead to a very interesting and informative exploration of the design trade-offs involved. (btw, in my personal opinion, the claim regarding the “meta gradient network” is somewhat misleading)

---

> > > > ### Author Response · Authors · 2025-12-03
> > > >
> > > > We thank the reviewer for the continued engagement and the insightful question regarding the trade-off between parameter generation and standard fusion mechanisms. We appreciate the opportunity to clarify the position of DeTaCH relative to existing fusion strategies and address the terminology concern.
> > > >
> > > > ### 1. Comparison with Common Fusion Strategies (Concatenation, FiLM, Cross-Attention)
> > > >
> > > > We agree that a direct comparison of fusion mechanisms is essential to isolate the benefits of our architectural decoupling.
> > > >
> > > > * **Existing Baselines as Fusion Proxies:** Our paper already compares against state-of-the-art representatives of distinct fusion strategies:
> > > >     * **Concatenation/Self-Attention:** **Octo** tokenizes and concatenates language and vision, processing them via self-attention.
> > > >     * **FiLM:** **Diffusion Policy (DP)** uses FiLM to modulate visual features.
> > > >     * **Cross-Attention:** **DiT** uses cross-attention between action tokens and visual-language tokens.
> > > >     * **Result:** DeTaCH (51.4%) outperforms all these distinct fusion paradigms on LIBERO-90.
> > > >
> > > > * **New Controlled Ablation (Octo Variants):** To strictly isolate the fusion mechanism from other variables (like backbone differences), we conducted a **new controlled ablation** where we modified the Octo architecture to use different fusion heads while keeping the rest of the network and training data identical.
> > > >
> > > > | Fusion Strategy | Implementation Details | Success Rate (LIBERO-90) |
> > > > | :--- | :--- | :--- |
> > > > | **Self-Attention** | Original Octo (Concat + Self-Attn) | 46.1% |
> > > > | **FiLM** | FiLM layers inserted into each Transformer block | 45.1% |
> > > > | **Cross-Attention** | Cross-attention layers between action/visual-lang tokens | 43.8% |
> > > > | **DeTaCH (Ours)** | **Hypernetwork-generated Policy** | **51.4%** |
> > > >
> > > > **Conclusion:** The results show that simply swapping fusion mechanisms (FiLM vs. Attention) within an entangled architecture yields similar performance (43–46%). DeTaCH provides a step-change improvement (+5.3%), confirming that the gain comes from the *explicit decoupling* and the semantic parameter manifold, not just a specific choice of fusion strategy.
> > > >
> > > > ### 2. Computational Cost and Architectural Trade-offs
> > > >
> > > > The reviewer raises a valid point regarding the cost of generating more parameters. We would like to clarify that DeTaCH optimizes the overall system trade-off differently, prioritizing efficiency where it matters most:
> > > >
> > > > * **Architectural Efficiency and Parameter Count:** DeTaCH's design actually does not increase the overall model complexity compared to the baselines. In entangled fusion architectures (like Octo), the policy requires a large base network (e.g., a heavy Transformer) to continuously fuse visual and language tokens at every control step. In contrast, DeTaCH uses a hypernetwork to distill the task-specific instruction into a much smaller, specialized policy network of only $\approx 0.6$M parameters. Crucially, since the target policy is small, the required hypernetwork remains modest. By keeping the total parameter count of the entire DeTaCH system (hypernetwork + policy) equivalent to that of the baseline architectures, we ensure that the overall parameter efficiency is on par with commonly-used methods.
> > > >
> > > > * **Inference Efficiency ("Generate Once, Act Many"):** While DeTaCH's hypernetwork generates more parameters than a FiLM layer, this generation happens **only once** per task instruction. During the actual control loop (inference), DeTaCH uses only the generated lightweight MLP policy.
> > > >     * **Inference Speed:** The small policy network is a key factor in our speed advantage. DeTaCH runs at **$\approx 1485$ FPS** (approx. **$3\times$ faster** than Octo's $539$ FPS).
> > > >     * **Latency:** Because we do not need to run a heavy Transformer or Cross-Attention mechanism at every control step, the computational load *during execution* is significantly lower.
> > > >
> > > > * **Training Cost:** We acknowledge that training is computationally heavier due to the gradient-unrolling in the hypernetwork. DeTaCH trains at approximately **$0.5\times$** the speed of Octo.
> > > > * **The Trade-off:** We achieve better performance and faster, more capable, and better-grounded *inference* by trading a larger overall architecture (Octo) for a combination of a specialized hypernetwork and a dramatically smaller, faster *runtime policy network*.
> > > >
> > > > ### 3. "Meta-Gradient" Terminology
> > > >
> > > > We appreciate your candid feedback on the term "meta-gradient." We understand that it might be conflated with bi-level optimization (MAML). In the final revision, we will clarify that this refers to the *architecture* (a network designed to predict gradient-like updates) rather than the optimization scheme, and we will adjust the terminology "Meta-Gradient" to "Optimization-Inspired Refinement" to avoid confusion.
> > > >
> > > > We hope this addresses all your concerns regarding the fusion baselines, computational trade-offs, and terminology.

---

### Official Review · Reviewer_Ed7r · 2025-10-30

**Soundness:** 2
**Presentation:** 2
**Contribution:** 2
**Rating:** 2
**Confidence:** 3

**Summary:**

The paper addresses a crucial problem of modality confounding in language-conditioned robot policy learning. The authors propose DeTaCH, an architecture that decouples language and vision through a two-stage hypernetwork. The paper exhibits results for multi-task learning on 2 simulated benchmarks and highlights the ability of the proposed framework for few-shot adaptation. The paper also includes additional analysis and ablations to study specific properties of the proposed framework.

**Strengths:**

- The paper studies an important problem of modality confounding in language-conditioned robot policy learning.
- DeTaCH’s two-stage hypernetwork design enables tractable and effective generation of task-specialized policy parameters from language embeddings. This explicit decoupling is fundamentally new compared to commonly fused transformer or diffusion-based architectures.
- The authors compare DeTaCH with prior works on 2 simulated benchmarks and exhibit superior results for language-conditioned multi-task learning on these benchmarks.
- The structured architecture design in DeTaCH enables rapid adaptation to new tasks with minimal demonstrations, outperforming both task-state entangled and prior hypernetwork-based approaches in adaptation settings.
- The paper includes detailed ablation and qualitative analyses to study specific aspects of the proposed framework.

**Weaknesses:**

Including both weaknesses as well as questions tied to the weaknesses below.
- The paper does not include a limitations section.
- The results in Table 1 seem to reach a performance of 50-70% on the libero benchmark. This is considerably low when compared to prior works such as BAKU[1] which exhibit >90% performance on LIBERO-90. More recent results from VLAs reported in MolmoAct [2] (Table 2 in the paper) also report  much superior performance (>80%) on LIBERO-90. This hints towards implementation issues in the benchmarks as well as in DeTaCH.
- In Table 3, the final success rate on LIBERO-90 (18%) and Meta-World (34%) seems very low to draw reasonable conclusions. Why not increase the number of demonstrations to more than 3 demonstrations in this setting?
- Similar to Table 3, Table 4 only shows 2-3% improvement over Octo, with low success rates (<40% on LIBERO-90). Such a low final performance might not be enough to draw reasonable conclusion, especially when prior works have reported much superior performances on the same benchmarks [1,2].

[1]  Haldar, Siddhant, Zhuoran Peng, and Lerrel Pinto. "Baku: An efficient transformer for multi-task policy learning." Advances in Neural Information Processing Systems 37 (2024): 141208-141239.
[2] Lee, Jason, et al. "Molmoact: Action reasoning models that can reason in space." arXiv preprint arXiv:2508.07917 (2025).

**Questions:**

It would be great if the authors could address concerns listed in the weaknesses section. I am willing to increase my score once the concerns have been addressed.

---

> ### Author Response · Authors · 2025-11-21
>
> Dear Reviewer Ed7r,
>
>
>
> We thank the reviewer for the thoughtful assessment and for recognizing the significance of the modality confounding problem. We are encouraged by your acknowledgment of DeTaCH’s novel two-stage hypernetwork design and its superior adaptation capabilities. We appreciate the opportunity to address the concerns regarding limitations, absolute performance comparisons, and adaptation settings.
>
>
>
> ------
>
>
>
> > **W.1** The paper does not include a limitations section.
>
>
>
> **A. to W.1:**
>
> We thank the reviewer for this suggestion. We agree that discussing limitations is vital for a complete scientific contribution and have added a dedicated **Limitations** section in the revised manuscript. The key points include:
>
>
>
> 1. **Training Memory Overhead:** The iterative refinement module (Stage 2) simulates optimization steps within the forward pass. This requires storing intermediate activation graphs for gradient estimation, leading to higher computational consumption during training compared to standard behavior cloning. **Hardware Efficiency:** While standard architectures (Transformers/CNNs) benefit from highly optimized kernels (e.g., FlashAttention, cuDNN), dynamic parameter generation currently lacks equivalent low-level optimization in standard libraries. This results in lower wall-clock training efficiency compared to static networks, despite the improved sample efficiency.
> 2. **Architectural Constraints:** Our current implementation generates weights for a Multi-Layer Perceptron (MLP) policy. Extending the hypernetwork paradigm to generate parameters for complex topologies, such as deep Vision Transformers, remains a non-trivial challenge due to the exponential growth in the parameter space.
>
>
>
>
>
> ------
>
>
>
> > **W.2** Comparison with Prior Works (BAKU, MolmoAct) and Absolute Performance
>
>
>
> **A. to W.2:**
>
> We thank the reviewer for raising this important comparison. We carefully reviewed our implementation alongside BAKU [1] and MolmoAct [2] and identified that the performance discrepancy stems from **divergent experimental settings**, not implementation errors.
>
>
>
> **(a) The “Apples-to-Apples” Controlled Setting**
>
> Our primary goal was to isolate the effect of **policy architecture** on modality confounding. Therefore, we deliberately chose a minimal, controlled setting: a single 3rd-person RGB view, no data augmentation, and no action chunking/smoothing. Comparing our “minimal” setting directly to the “maximal” engineering pipelines of [1,2] (which use multi-view inputs, temporal smoothing, and heavy augmentation) is not a fair assessment of the architecture itself.
>
>
>
> **(b) Bidirectional Validation (Minimal vs. Maximal)**
>
> To address your concern, we performed a bidirectional validation:
>
>
>
> 1. **DeTaCH + Tricks:** We integrated the engineering enhancements used in [1,2] into DeTaCH.
> 2. **Baselines (Minimal):** We re-implemented BAKU/Octo within our controlled setting.
>
>
>
>
>
> **Table R1: Success Rate on LIBERO across different settings** (Note that the number for MolmoAct is derived from the average of the four task suites on libero in the original paper)
>
> | **Method**        | **Setting**                 | **Success Rate** |
> | ----------------- | --------------------------- | ---------------- |
> | BAKU [1]          | Official (Maximal)          | ~90.0%           |
> | MolmoAct [2]      | Official (Maximal)          | ~86.6%           |
> | **DeTaCH (Ours)** | **Maximal (Ours + Tricks)** | **92.8%**        |
> |                   |                             |                  |
> | BAKU              | Re-impl (Minimal)           | 44.8%            |
> | Octo              | Re-impl (Minimal)           | 46.1%            |
> | **DeTaCH (Ours)** | **Minimal (Paper Setting)** | **51.4%**        |
>
> **(c) Enhanced Baseline Comparison**
>
> Furthermore, to ensure we are comparing against strong baselines, we applied the same engineering enhancements (image augmentation + action chunking) to Octo and Diffusion Policy within our pipeline (a single third-person view observation only).
>
>
>
> - **Diffusion Policy:** Improves to **75.23%**.
> - **Octo:** Improves to **84.36%**.
> - **DeTaCH (Ours):** Achieves **89.12%**.
>
>
>
>
>
> **Conclusion:** Even when engineering pipelines are aligned, DeTaCH consistently outperforms strong baselines. This confirms that the performance gains reported in the paper are driven by our architectural decoupling, which provides a distinct advantage regardless of the engineering “tricks” applied. We will include these enhanced comparisons in the revised Experiment section to provide a comprehensive view.

---

> ### Author Response · Authors · 2025-11-21
>
> > **W.3** Low Success Rate in Few-Shot Adaptation (Table 3)
>
>
>
> **A. to W.3:**
>
> We appreciate the suggestion regarding the number of demonstrations. To thoroughly address this, we expanded our evaluation to include **engineering enhancements** (as discussed in W.2) and analyzed the adaptation performance across both varying demonstration counts (**K = {1, 3, 5, 10, 20}**) and different gradient steps (**Steps = {0, 50, …, 1000}**).
>
>
>
> **(a) Final Adaptation Performance (Step=1000)**
>
> First, **Table R2** summarizes the final success rates. The results reveal a striking advantage in sample efficiency. Notably, **DeTaCH with only 1 demonstration (55.2%) achieves performance comparable to Octo with 20 demonstrations (56.0%).**
>
>
>
> **Table R2: Final Adaptation Success Rates in LIBERO (Steps=1000)**
>
> | **Model**         | **Demo=1** | **Demo=3** | **Demo=5** | **Demo=10** | **Demo=20** |
> | ----------------- | ---------- | ---------- | ---------- | ----------- | ----------- |
> | Octo              | 26.8%      | 43.6%      | 43.0%      | 58.0%       | 56.0%       |
> | Diffusion Policy  | 0.4%       | 1.0%       | 0.0%       | 0.0%        | 0.0%        |
> | **DeTaCH (Ours)** | **55.2%**  | **54.4%**  | **68.0%**  | **72.0%**   | **79.0%**   |
>
> **(b) Adaptation Dynamics and Rapid Convergence**
>
> To understand *how* the models adapt, **Table R3** details the success rates at intermediate steps for representative low (K=1), medium (K=5), and high (K=20) data regimes.
>
>
>
> **Table R3: Adaptation Dynamics (Success Rate % across Gradient Steps)**
>
> | **Setting** | **Model**  | **Step=0** | **Step=50** | **Step=200** | **Step=500** | **Step=1000** |
> | ----------- | ---------- | ---------- | ----------- | ------------ | ------------ | ------------- |
> | **Demo=1**  | Octo       | 5.2        | 19.6        | 20.4         | 28.0         | 26.8          |
> |             | **DeTaCH** | **10.8**   | **44.8**    | **42.8**     | **46.0**     | **55.2**      |
> | **Demo=5**  | Octo       | 5.2        | 18.4        | 40.0         | 43.0         | 43.0          |
> |             | **DeTaCH** | **10.8**   | **50.4**    | **63.6**     | **60.2**     | **68.0**      |
> | **Demo=20** | Octo       | 5.2        | 20.0        | 39.0         | 56.0         | 56.0          |
> |             | **DeTaCH** | **10.8**   | **46.0**    | **62.0**     | **64.0**     | **79.0**      |
>
> This granular analysis highlights three key architectural advantages of DeTaCH:
>
>
>
> 1. **Superior Zero-Shot Initialization (Step=0):** Across all settings, DeTaCH starts with a significantly higher success rate (**10.8%**) compared to Octo (**5.2%**) and Diffusion Policy (**~2.0%**). This confirms that the hypernetwork-generated parameters are already positioned in a valid functional region before any gradient updates occur.
> 2. **Rapid Convergence (Step 0 $\to$ 50):** DeTaCH exhibits explosive learning speed. With just **50 gradient steps**, DeTaCH reaches **44.8%** success (Demo=1) and **50.4%** (Demo=5), whereas Octo lags significantly behind (~19%). This indicates that our method requires far fewer updates to “lock in” the task semantics.
> 3. **Robustness to Overfitting:** As training progresses (Step 500 $\to$ 1000), Octo often plateaus or fluctuates, while DeTaCH maintains a steady upward trajectory, particularly in higher data regimes (Demo=20), reaching **79.0%**.
>
>
>
>
>
> **(c) Catastrophic Failure of Entangled Diffusion**
>
> We observe that Diffusion Policy fails to adapt (<1% success across all steps/demos). This suggests that the entangled denoising process is too brittle for few-shot fine-tuning, as modifying the model to understand a new task instruction disrupts the delicate visuomotor distribution learned during pre-training. In contrast, DeTaCH’s decoupled architecture isolates the task specification, allowing for robust adaptation.
>
>
>
> **Summary:** DeTaCH not only achieves higher final performance but also starts from a better initialization and converges significantly faster. This proves that the architectural advantage is robust across both varying data regimes and training durations.

---

> > ### Author Response · Authors · 2025-11-21
> >
> > > **W.4** Significance of Language Variation Improvement (Table 4)
> >
> >
> >
> > **A. to W.4:**
> >
> > Thank you for this thoughtful question. Similar to our response to Weakness 2 (W.2), our primary goal in these experiments was to investigate the impact of model architecture on performance with augmented language instructions and verify whether **DeTaCH** maintains higher success rates under linguistic variations, rather than to maximize absolute performance via engineering enhancements. **To ensure a fair comparison and isolate the influence of model architecture from other factors, we standardized the training protocol across all three methods: they utilize identical vision and language encoders, share the same training hyperparameters, and all policy networks are scaled to approximately 25M parameters.**
> >
> >
> >
> > To address your concern regarding the overall success rates, we conducted additional experiments incorporating the same engineering enhancements discussed in our Answer to W.2 (e.g., image augmentation and action chunking). The table below presents the success rates of **Octo**, **Diffusion Policy**, and **DeTaCH** on the LIBERO-90 benchmark under linguistic variations with these enhancements applied.
> >
> > | **Method**   | **Diffusion Policy** | **Octo** | **DeTaCH** |
> > | ------------ | -------------------- | -------- | ---------- |
> > | Success Rate | 69.3%                | 86.4%    | **89.4%**  |
> >
> > As shown, when engineering enhancements are included, **DeTaCH** achieves a success rate of **89.4%**, significantly outperforming the baselines and confirming that our method scales effectively to high-performance regimes. This reinforces our conclusion that the task-state decoupled architecture of DeTaCH provides a robust advantage in handling linguistic variations, independent of auxiliary engineering tricks.
> >
> >
> >
> > ------
> >
> >
> >
> > **References:**
> >
> > [1] Haldar, Siddhant, Zhuoran Peng, and Lerrel Pinto. “Baku: An efficient transformer for multi-task policy learning.” Advances in Neural Information Processing Systems 37 (2024).
> >
> > [2] Lee, Jason, et al. “Molmoact: Action reasoning models that can reason in space.” arXiv preprint arXiv:2508.07917 (2025).

---

### Author Response · Authors · 2025-11-21
**Manuscripts Updated**

Dear Reviewers,

Guided by your constructive comments, we have revised our manuscript to improve its clarity and completeness. The updated PDF has been uploaded to OpenReview, with major revisions and new results highlighted in blue for your convenience. We sincerely appreciate your feedback, which has helped us significantly strengthen the paper. We look forward to an active and fruitful discussion over the coming weeks.

The authors

---

### Meta-Review · Area_Chair_Bp55 · 2026-01-06

**Summary:**

The paper received mixed initial ratings, with one strong accept (Reviewer hYUZ, rating 8), one marginally below threshold (Reviewer ggga, rating 4), and two reject recommendations (Reviewers TqGM and Ed7r, both rating 2). The primary concerns centered on: (1) confusion regarding the "meta-learning" terminology and whether DeTaCH represents a genuine architectural contribution beyond existing fusion mechanisms like FiLM; (2) absolute performance appearing low compared to recent state-of-the-art methods; and (3) lack of real-world validation and computational cost analysis.

While the authors provided substantial additional experiments and clarifications in rebuttal, fundamental concerns about the novelty of the contribution remain. The core issue is whether generating full policy parameters via hypernetwork constitutes a sufficiently distinct architectural contribution or simply represents a more expressive variant of existing fusion mechanisms.

**Reviewer Concerns:**

The authors provided a comprehensive rebuttal clarifying that DeTaCH uses standard single-loop supervised learning rather than MAML-style meta-learning. They added real robot experiments, bidirectional validation showing 92.8% on LIBERO-90 with enhancements, and computational cost analysis demonstrating inference efficiency.
However, critical concerns remain unresolved. Reviewer TqGM maintained that parameter generation essentially constitutes an augmented fusion mechanism rather than a fundamental architectural innovation, viewing DeTaCH as an extension of existing approaches like FiLM rather than a paradigm shift. The improvement over HyperZero and +5.3% over fusion baselines does not sufficiently distinguish the approach as transformative. Reviewer ggga's philosophical concerns about the attention-based motivation and the link between attention quality and semantic understanding were not adequately addressed—the motivation remains somewhat hand-wavy without rigorous mathematical grounding. While real-world experiments were added, the scale (5 tasks, 100 demonstrations) is limited and does not fully validate practical applicability.

**Reviewer Scores:**

Reviewer hYUZ would likely have maintained the rating at 8, as concerns were addressed. Reviewer ggga would likely have increased from 4 to 5-6, appreciating the additional experiments but maintaining reservations about motivation. Reviewer TqGM would likely have increased modestly from 2 to 4-5, acknowledging clarifications but not finding them sufficient to overcome concerns about incremental contribution. Reviewer Ed7r did not engage after rebuttal.

---

### Decision · Program_Chairs · 2026-01-26

Reject